# Improving Generalization with Flat Hilbert Bayesian Inference

Tuan Truong [* 1]   Quyen Tran [* 1]   Ngoc-Quan Pham [1]   Nhat Ho [2]   Dinh Phung [1 3]   Trung Le [3]

## Abstract

We introduce Flat Hilbert Bayesian Inference (FHBI), an algorithm designed to enhance generalization in Bayesian inference. Our approach involves an iterative two-step procedure with an adversarial functional perturbation step and a functional descent step within a reproducing kernel Hilbert space. This methodology is supported by a theoretical analysis that extends previous findings on generalization ability from finite-dimensional Euclidean spaces to infinite-dimensional functional spaces. To evaluate the effectiveness of FHBI, we conduct comprehensive comparisons against nine baseline methods on the `VTAB-1K` benchmark, which encompasses 19 diverse datasets across various domains with diverse semantics. Empirical results demonstrate that FHBI consistently outperforms the baselines by notable margins, highlighting its practical efficacy.

## 1. Introduction

Quantifying and tackling uncertainty in deep learning is one of the most challenging problems, mainly due to the inherent randomness of the real world and the presence of noisy data. Bayesian inference provides a robust framework for understanding complex data, allowing for probabilistic interpretation of deep learning models and reasoning under uncertainty. This approach not only facilitates predictions but also enables the quantification of uncertainty. A primary challenge in this domain is the computation and sampling from intricate distributions, mainly when dealing with deep learning models. One effective strategy to tackle this issue is variational inference, which seeks to approximate the true posterior distribution with simpler forms, known as approximate posteriors, while optimizing a variational lower bound. Several techniques have been developed in this area, including those by Kingma & Welling (2014); Kingma et al. (2015), and Blundell et al. (2015), who extended the Gaussian variational posterior approximation for neural networks, as well as Gupta & Nagar (2018), who enhanced the flexibility of posterior approximations. In addition to variational methods, various particle sampling techniques have been proposed for Bayesian inference, especially in scenarios requiring multiple models. Notable particle sampling methods include Hamiltonian Monte Carlo (HMC) (Neal, 1996), Stochastic Gradient Langevin Dynamics (SGLD) (Welling & Teh, 2011), Stochastic Gradient HMC (SGHMC) (Chen et al., 2014), and Stein Variational Gradient Descent (SVGD) (Liu & Wang, 2016). Each method contributes to a deeper understanding and more practical application of Bayesian inference in deep learning.

Besides quantifying uncertainty, tackling overfitting is a significant challenge in machine learning. Overfitting often occurs when the training process gets stuck in local minima, leading to a model that fails to generalize well to unseen data. This problem is mainly due to loss functions' high-dimensional and non-convex nature, which often exhibit multiple local minima in the loss landscape. In standard deep network training, flat minimizers improve model generalization (Keskar et al., 2017; Kaddour et al., 2022; Li et al., 2022). Among the flat minimizers, Sharpness-Aware Minimization (SAM) (Foret et al., 2021) has emerged as a practical approach by concurrently minimizing the empirical loss and reducing the sharpness of the loss function. Recently, SAM has demonstrated its versatility and effectiveness across a wide range of tasks, including meta-learning (Abbas et al., 2022), vision models (Chen et al., 2021), and language models (Bahri et al., 2022).

**Contribution.** We combine flat minimizers and particle-based Bayesian methods to introduce a novel Bayesian inference framework with improved generalization ability. To accomplish this, we first present Theorem 4.2, which strengthens prior generalization bounds from *finite-dimensional* Euclidean spaces to the reproducing kernel Hilbert spaces (RKHS), which are broader functional spaces and are typically *infinite-dimensional*. Notably, this theorem introduces the notion of *functional sharpness* that offers an insight to improve the generalization ability of current particle-sampling methods. Subsequently, Theorem 4.3 translates

---

[*]Equal contribution [1]Movian AI, Vietnam [2]The University of Texas at Austin, USA [3]Monash University, Australia. Correspondence to: Trung Le <trunglm@monash.edu>.

*Proceedings of the 42$^{nd}$ International Conference on Machine Learning*, Vancouver, Canada. PMLR 267, 2025. Copyright 2025 by the author(s).

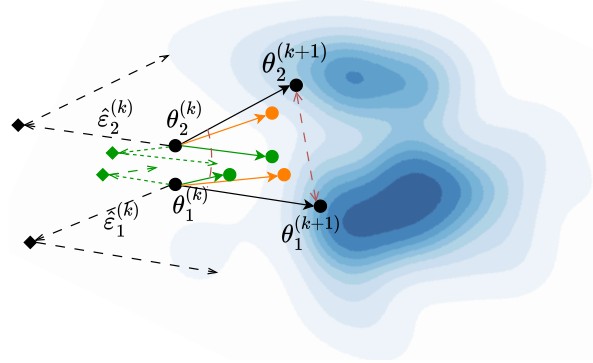

Figure 1. Schematic of SAM w. independent particles (green), SVGD (orange), and our FHBI (black) updates. SAM's particles are not aware of others' trajectories. SVGD only seeks the modes and promotes *spatial* diversity. FHBI seeks the modes, minimizes sharpness, and promotes *spatial* and *angular* diversity.

these notions of functional sharpness and generalization from the context of RKHS to the context of Bayesian inference. This analysis establishes a connection between the population and empirical KL loss, providing a strategy to enhance generalization by minimizing the population KL loss. Motivated by these two theorems, we present Flat Hilbert Bayesian Inference (FBVI), a practical algorithm that employs a dual-step *functional sharpness-aware* update procedure in RKHS. This approach improves the generalization of sampled particles, thereby enhancing the quality of the ensemble. Overall, our contributions are as follows:

1. We present a theoretical analysis that characterizes generalization ability over the functional space. This analysis generalizes prior works from the Euclidean space to infinite-dimensional functional space, thereby introducing the notion of *functional sharpness* i.e., the sharpness of the functional spaces.

2. Building on this theoretical foundation, we propose a practical particle-sampling algorithm that enhances the generalization ability over existing methods. We conducted extensive experiments comparing our Flat Hilbert Bayesian Inference (FHBI) algorithm with nine baselines on the `VTAB-1K` benchmark, which includes 19 datasets across various domains and semantics. Experimental results demonstrated that our algorithm outperforms these baselines by notable margins.

**Organization.** The paper is structured as follows: Section 2 reviews the related works on Bayesian inference and the development of flat minimizers. Section 3 provides the necessary background and notations. Section 4 discusses the motivation and theoretical development behind our sharpness-aware particle-sampling approach. Section 5 presents experimental results, comparing our algorithm

against various Bayesian inference baselines across diverse settings. Section 6 offers a deeper analysis of FHBI's behavior to gain further insight into its effectiveness over the baselines.

## 2. Related Works

**Sharpness-aware minimization.** Flat minimizers are more robust to the shifts between training and test losses, thereby enhancing the generalization ability of neural networks (Jiang et al., 2020; Petzka et al., 2021; Dziugaite & Roy, 2017). The relationship between generalization and the width of minima has been studied both theoretically and empirically in several prior works (Hochreiter & Schmidhuber, 1994; Neyshabur et al., 2017; Dinh et al., 2017; Fort & Ganguli, 2019). Consequently, a variety of methods have been developed to search for flat minima (Pereyra et al., 2017; Chaudhari et al., 2017; Keskar et al., 2017; Izmailov et al., 2018). Among the flat minimizers, Sharpness-Aware Minimization (SAM) (Foret et al., 2021) has gained significant attention due to its effectiveness. SAM has been leveraged across a wide range of tasks and domains, including domain generalization (Cha et al., 2021; Wang et al., 2023; Zhang et al., 2023), federated learning (Caldarola et al., 2022; Qu et al., 2022), Bayesian networks (Nguyen et al., 2023; Möllenhoff & Khan, 2023), and meta-learning (Abbas et al., 2022). Moreover, SAM has demonstrated its ability to enhance generalization in both vision models (Chen et al., 2021) and language models (Bahri et al., 2022).

Nevertheless, these studies are constrained to finite-dimensional Euclidean spaces. In this work, we strengthen these generalization principles to infinite-dimensional functional spaces and propose a particle-sampling method grounded in this theoretical framework.

**Bayesian Inference.** Two main strategies were widely employed in the literature of Bayesian inference. The first paradigm is *Variational Inference*, which aims to approximate a target distribution by selecting a distribution from a family of potential approximations and optimizing a variational lower bound. Graves (2011) introduced the use of a Gaussian variational posterior approximation for neural network weights, which was later extended by Kingma & Welling (2014); Kingma et al. (2015); Blundell et al. (2015) with the reparameterization trick to facilitate training deep latent variable models. Louizos & Welling (2017) proposed using a matrix-variate Gaussian to model entire weight matrices (Gupta & Nagar, 2018) to increase further the flexibility of posterior approximations, which offers a novel approach to approximate the posterior. Subsequently, various alternative structured forms of the variational Gaussian posterior were proposed, including the Kronecker-factored approximations (Zhang et al., 2018; Ritter et al., 2018; Rossi

et al., 2020), or non-centered or rank-1 parameterizations (Ghosh et al., 2018; Dusenberry et al., 2020). Recently, Pham et al. (2025) proposes to incorporate sharpness and distributional robustness to Bayesian Inference, thereby modelling the interactions between particles and allowing a more diverse set of model particles.

The second paradigm of Bayesian inference is *Markov Chain Monte Carlo* (MCMC), which involves sampling multiple models from the posterior distribution. MCMC has been applied to neural network inference, such as Hamiltonian Monte Carlo (HMC) (Neal, 1996). However, HMC requires the computation of full gradients, which can be computationally expensive. To address this, Stochastic Gradient Langevin Dynamics (SGLD) (Welling & Teh, 2011) integrates first-order Langevin dynamics within a stochastic gradient framework. Stochastic Gradient HMC (SGHMC) (Chen et al., 2014) further incorporates stochastic gradients into Bayesian inference, enabling scalability and efficient exploration of different solutions. Another critical approach, Stein Variational Gradient Descent (SVGD) (Liu & Wang, 2016), closely related to our work, uses a set of particles that converge to the target distribution. It is also theoretically established that SGHMC, SGLD, and SVGD asymptotically sample from the posterior as the step sizes approach zero.

## 3. Backgrounds and Notations

**Bayesian Inference.** Consider a family of neural networks $f_{\boldsymbol{\theta}}(x)$, where the random variable $\boldsymbol{\theta}$ represents the model parameters and takes values in the model space $\Theta \subset \mathbb{R}^d$. We are given a training set $\mathcal{S} = \{(x_i, y_i)\}_{i=1}^n$ of $n$ i.i.d observations from the data space $\mathcal{X} \times \mathcal{Y}$, and the prior distribution of the parameters $p(\boldsymbol{\theta})$. In the literature on Bayesian inference problems, prior works typically focus on approximating the *empirical posterior* $\mathbb{P}_{\boldsymbol{\theta}|\mathcal{S}}$, whose density function $p(\boldsymbol{\theta}|\mathcal{S})$ is defined as:

$$p(\boldsymbol{\theta}|\mathcal{S}) \propto p(\boldsymbol{\theta}) \prod_{i=1}^n p(y_i|x_i, \mathcal{S}, \boldsymbol{\theta}),$$

where the prior distribution $\mathbb{P}_{\theta}$ has the density function $p(\boldsymbol{\theta})$. The likelihood term is proportional to

$$p(y|x, \mathcal{S}, \boldsymbol{\theta}) \propto \exp\left( -\frac{1}{n}\ell(f_{\boldsymbol{\theta}}(x), y) \right),$$

with some loss function $\ell$, such as the CrossEntropy loss, and a sufficiently expressive model $f_{\theta}$. Then, the empirical posterior is:

$$p(\boldsymbol{\theta}|\mathcal{S}) \propto \exp\left( -\frac{1}{n}\sum_{i=1}^n \ell(f_{\boldsymbol{\theta}}(x_i), y_i) \right)p(\boldsymbol{\theta}). \quad (1)$$

More formally, the empirical posterior is equal to:

$$p(\boldsymbol{\theta}|\mathcal{S}) = \exp\left( -\frac{1}{n}\sum_{i=1}^n \ell(f_{\boldsymbol{\theta}}(x_i), y_i) \right)p(\boldsymbol{\theta})/Z_{\mathcal{S}}, \quad (2)$$

where $Z_{\mathcal{S}}$ is the normalizing constant. We define the population and empirical losses as follows:

$$\mathcal{L}_{\mathcal{D}}(\boldsymbol{\theta}) = \mathbb{E}_{(x,y)\sim\mathcal{D}}[\ell(f_{\boldsymbol{\theta}}(x), y)],$$

$$\mathcal{L}_{\mathcal{S}}(\boldsymbol{\theta}) = \mathbb{E}_{(x,y)\sim\mathcal{S}}[\ell(f_{\boldsymbol{\theta}}(x), y)] = \frac{1}{n}\sum_{i=1}^n \ell(f_{\boldsymbol{\theta}}(x_i), y_i).$$

The *population loss* is defined as the expected loss over the entire data-label distribution. In contrast, the *empirical loss* is the average loss computed over a given training set $\mathcal{S}$. Based on these definitions, the empirical posterior in Eq. 2 can be written as:

$$p(\boldsymbol{\theta}|\mathcal{S}) = \exp(-\mathcal{L}_{\mathcal{S}}(\boldsymbol{\theta}))p(\boldsymbol{\theta})/Z_{\mathcal{S}}.$$

Intuitively, models with parameters $\boldsymbol{\theta}$ that fit well to the training set $\mathcal{S}$ lead to lower empirical loss values, resulting in higher density in the empirical posterior. This formulation has appeared in prior works on Bayesian Inference, such as (Nguyen et al., 2023). However, simply fitting to the training samples can lead to overfitting. To improve generalization, we are more concerned with performance over the entire data distribution $\mathcal{D}$ rather than just the specific sample $\mathcal{S}$. Accordingly, we define the population posterior as $\mathbb{P}_{\mathcal{D}}$ whose density is given by:

$$p(\boldsymbol{\theta}|\mathcal{D}) = \exp(-\mathcal{L}_{\mathcal{D}}(\boldsymbol{\theta}))p(\boldsymbol{\theta})/Z_{\mathcal{D}}, \quad (3)$$

with the normalizing constant $Z_{\mathcal{D}}$. This population posterior is more general than the empirical posterior, as it captures the true posterior of the parameters under the full data distribution. However, understanding the population posterior is particularly challenging because we can only access the empirical loss $\mathcal{L}_{\mathcal{S}}(\boldsymbol{\theta})$, not the population loss $\mathcal{L}_{\mathcal{D}}(\boldsymbol{\theta})$. In this paper, we deviate from prior approaches that primarily focus on approximating the empirical posterior and instead propose a particle-sampling method to approximate the population posterior.

**Reproducing Kernel Hilbert Space (RKHS).** Let $k(\boldsymbol{\theta}, \boldsymbol{\theta}') : \Theta \times \Theta \to \mathbb{R}$ be a positive definite kernel operating on the model space. The reproducing kernel Hilbert space (RKHS) $\mathcal{H}$ of $k(\boldsymbol{\theta}, \boldsymbol{\theta}')$ is the closure of the linear span $\{f : f(\cdot) = \sum_i a_i k(\cdot, \boldsymbol{\theta}_i), a_i \in \mathbb{R}, \boldsymbol{\theta}_i \in \Theta\}$. For $f(\boldsymbol{\theta}) = \sum_i a_i k(\boldsymbol{\theta}, \boldsymbol{\theta}_i)$ and $g(\boldsymbol{\theta}) = \sum_j b_j k(\boldsymbol{\theta}, \boldsymbol{\theta}_j)$, $\mathcal{H}$ is equipped with the inner product defined by $\langle f, g \rangle_{\mathcal{H}} = \sum_{ij} a_i b_j k(\boldsymbol{\theta}_i, \boldsymbol{\theta}_j)$. For all $\boldsymbol{\theta} \in \Theta$, there exists a unique

element $K_{\boldsymbol{\theta}} \in \mathcal{H}$ with the reproducing property that $f(\boldsymbol{\theta}) = \langle f, K_{\boldsymbol{\theta}} \rangle_{\mathcal{H}}$ for any $f \in \mathcal{H}$.

Given that $\mathcal{H}$ is a scalar-valued RKHS with kernel $k(\theta, \theta')$, $\mathcal{H}^d = \mathcal{H} \times \mathcal{H} \times \cdots \times \mathcal{H}$ is a vector-valued RKHS of functions $\boldsymbol{f} = [f_1, f_2, \cdots, f_d]$ corresponding to the kernel $K(\boldsymbol{\theta}, \boldsymbol{\theta}') = k(\theta, \theta')\boldsymbol{I}$. $\mathcal{H}^d$ is equipped with the inner product $\langle \boldsymbol{f}, \boldsymbol{g} \rangle_{\mathcal{H}^d} = \sum_{i=1}^{d} \langle f_i, g_i \rangle_{\mathcal{H}}$.

Let $\boldsymbol{F}[\boldsymbol{f}]$ be a functional on $\boldsymbol{f} \in \mathcal{H}^d$. Similar to the definition by (Liu & Wang, 2016), the (functional) gradient of $\boldsymbol{F}$ is defined as a function $\nabla_{\boldsymbol{f}} \boldsymbol{F}[\boldsymbol{f}] \in \mathcal{H}^d$ such that for any $\boldsymbol{g} \in \mathcal{H}^d$ and $\epsilon \in \mathbb{R}$

$$\boldsymbol{F}[\boldsymbol{f} + \epsilon \boldsymbol{g}] = \boldsymbol{F}[\boldsymbol{f}] + \epsilon \langle \nabla_{\boldsymbol{f}} \boldsymbol{F}[\boldsymbol{f}], g \rangle_{\mathcal{H}^d} + \mathcal{O}(\epsilon^2). \quad (4)$$

**Stein Variational Gradient Descent (SVGD).** Given a general target distribution $p(\boldsymbol{\theta})$, SVGD (Liu & Wang, 2016) aims to find a flow of distributions $\{q^{(k)}\}_k$ that minimizes the KL distance to the target distribution. Motivated by the Stein identity and Kernelized Stein Discrepancy, SVGD proposes the update $q^{(k+1)} = q^{(k)}_{[\boldsymbol{T}]}$, in which $\boldsymbol{T} : \Theta \to \Theta$ is a smooth one-to-one push-forward map of the form $\boldsymbol{T}(\boldsymbol{\theta}) = \boldsymbol{\theta} + \epsilon \phi^*_{p,q}(\boldsymbol{\theta})$ in which:

$$\phi^*_{p,q}(\cdot) = \mathbb{E}_{\boldsymbol{\theta} \sim q}[\mathcal{A}_p k(\boldsymbol{\theta}, \cdot)]$$
$$\mathcal{A}_p \phi(\boldsymbol{\theta}) = \phi(\boldsymbol{\theta}) \nabla_{\boldsymbol{\theta}} \log p(\boldsymbol{\theta})^\top + \nabla_{\boldsymbol{\theta}} \phi(\boldsymbol{\theta}).$$

$\mathcal{A}_p$ is known as the Stein operator, which acts on $\phi$ and produces a zero-mean function $\mathcal{A}_p \phi(\boldsymbol{\theta})$ when $\boldsymbol{\theta} \sim p$. While SVGD is designed for general target distributions $p$, in the context of Bayesian inference, it is only applicable to the empirical posterior rather than the population posterior, which we will discuss in detail in the next section.

## 4. Flat Hilbert Bayesian Inference (FHBI)

Consider the Bayesian inference problem of approximating a posterior distribution. In prior works, such as SVGD (Liu & Wang, 2016), when applying to the context of Bayesian inference, the methods are only applicable to the *empirical posterior* $p(\boldsymbol{\theta}|\mathcal{S})$ because we only have access to the empirical loss. It is evident that when sampling a set of $m$ particle models $\boldsymbol{\theta}_{1:m}$ from $p(\boldsymbol{\theta}|\mathcal{S})$, these particles congregate in the high-density regions of the empirical posterior $p(\boldsymbol{\theta}|\mathcal{S})$, corresponding to the areas with low *empirical loss* $\mathcal{L}_{\mathcal{S}}(\boldsymbol{\theta})$. However, to avoid overfitting, it is preferable to sample the particle models $\boldsymbol{\theta}_{1:m}$ from the *population posterior* $p(\boldsymbol{\theta}|\mathcal{D}) \propto \exp(-\mathcal{L}_{\mathcal{D}}(\boldsymbol{\theta}))p(\boldsymbol{\theta})$, as this approach directs the particle models $\boldsymbol{\theta}_{1:m}$ towards regions with low values of the *population loss* $\mathcal{L}_{\mathcal{D}}(\boldsymbol{\theta})$, thus improving generalization ability. To better understand this motivation from a theoretical perspective, consider the following proposition, with the proof provided in Appendix A.1:

**Proposition 4.1.** *Consider the problem of finding the distribution $\mathbb{Q}$ that solves:*

$$\mathbb{Q}^* = \min_{\mathbb{Q} \ll \mathbb{P}_\theta} \left\{ \mathbb{E}_{\theta \sim \mathbb{Q}}[\mathcal{L}_{\mathcal{D}}(\boldsymbol{\theta})] + D_{\mathrm{KL}}(\mathbb{Q} \| \mathbb{P}_\theta) \right\} \quad (5)$$

*where we search over $\mathbb{Q}$ absolutely continuous w.r.t $\mathbb{P}_{\boldsymbol{\theta}}$, and the second term is the regularization term. The closed-form solution to this problem is exactly the population posterior defined in Eq. 3.*

In this proposition, we aim to identify the posterior distribution that minimizes the *expected population loss*, where the expectation is taken over the entire parameter space with $\boldsymbol{\theta} \sim \mathbb{Q}^*$ while maintaining proximity to the prior distribution to ensure simplicity. With access to this posterior $\mathbb{Q}^*$, we can sample a set of particles whose average performance optimally minimizes the population loss. Since the solution to this optimization problem corresponds precisely to the population posterior, the *ensemble of the particles* sampled from $\mathbb{Q}^* \equiv p(\theta|\mathcal{D})$ effectively minimizes the average value of the population loss. This is because $\mathbb{Q}^*$ is explicitly chosen to minimize the expected value of the population loss $\mathcal{L}_{\mathcal{D}}$, which means the ensemble fits the whole data distribution instead of overfitting to the specific dataset $\mathcal{S}$, therefore establishes improved generalizability. Consequently, this proposition theoretically asserts that sampling from $p(\boldsymbol{\theta}|\mathcal{D})$ enhances the generalizability of the ensemble.

### 4.1. Theoretical analysis

Motivated by this observation, we advance prior work by *approximating the population posterior*. Specifically, to improve generalizability, our objective is to approximate the target population posterior distribution $p(\boldsymbol{\theta}|\mathcal{D})$ using a simpler distribution $q^*(\boldsymbol{\theta})$ drawn from a predefined set of distributions $\mathcal{F}$. This is achieved by minimizing the KL divergence:

$$q^* = \arg\min_{q \in \mathcal{F}} D_{\mathrm{KL}}\left( q(\boldsymbol{\theta}) \| p(\boldsymbol{\theta}|\mathcal{D}) \right). \quad (6)$$

Ideally, the set $\mathcal{F}$ should be simple enough for a simple solution and effective inference while sufficiently broad to approximate a wide range of target distributions closely. Let $q(\boldsymbol{\theta})$ be the density of a reference distribution. We define $\mathcal{F}$ as the set of distributions for random variables of the form $\vartheta = \boldsymbol{T}(\boldsymbol{\theta})$, where $\boldsymbol{T} : \Theta \to \Theta$ is a smooth, bijective mapping, and $\boldsymbol{\theta}$ is sampled from $q$. By variable change, the density of $\vartheta$, denoted as $q_{[\boldsymbol{T}]}(\cdot)$, is expressed as follows:

$$q_{[\boldsymbol{T}]}(\vartheta) = q(\boldsymbol{T}^{-1}(\vartheta)) |\det(\nabla_{\vartheta} \boldsymbol{T}^{-1}(\vartheta))|.$$

We restrict the set of the smooth transformations $\boldsymbol{T}$ to the set of push-forward maps of the form $\boldsymbol{T}(\boldsymbol{\theta}) = \boldsymbol{\theta} + \boldsymbol{f}(\boldsymbol{\theta})$, where

$f \in \mathcal{H}^d$. When $\|f\|_{\mathcal{H}^d}$ is sufficiently small, the Jacobian of $T = I + f$ is full-rank where $I$ denotes the identity map, in which case $T$ is guaranteed to be a one-to-one map according to the inverse function theorem. Under this restriction, the problem is equivalent to solving an optimization problem over the RKHS:

$$f^* = \underset{f \in \mathcal{H}^d, \|f\|_{\mathcal{H}^d} \leq \epsilon}{\arg\min} D_{\mathrm{KL}}\left(q_{[I+f]}(\theta)\|p(\theta|\mathcal{D})\right).$$

The challenge with this optimization problem lies in our lack of access to the population loss function $\mathcal{L}_{\mathcal{D}}(\theta)$ and the population posterior distribution $p(\theta|\mathcal{D})$. We present our first theorem to address this issue, which characterizes generalization ability in the functional space $\mathcal{H}^d$. The proof of this theorem can be found in Appendix A.2.

**Theorem 4.2** (Informal). *Let $\tilde{\ell} : \mathcal{H}^d \times \mathcal{X} \times \mathcal{Y} \to \mathbb{R}^+$ be a loss function on the RKHS $\mathcal{H}^d$ and the data space. Define $\tilde{L}_{\mathcal{D}}(f) = \mathbb{E}_{(x,y)\sim\mathcal{D}}[\tilde{\ell}(f,x,y)]$ and $\tilde{L}_{\mathcal{S}}(f) = \frac{1}{n}\sum_{i=1}^{n} \tilde{\ell}(f,x_i,y_i)$ be the corresponding population and empirical losses. Then for any $\rho > 0$ and any distribution $\mathcal{D}$, with probability of $1 - \delta$ over the choice of the training set $\mathcal{S} \sim \mathcal{D}^n$, we have:*

$$\tilde{L}_{\mathcal{D}}(f) \leq \max_{f' \in \mathcal{H}^d, \|f'-f\|_{\mathcal{H}^d} \leq \rho} \tilde{L}_{\mathcal{S}}(f')$$
$$+ \mathcal{O}\left(\sqrt{\frac{\log(1 + \frac{1}{\rho^2}) + \log\left(\frac{n}{\delta}\right)}{n-1}}\right),$$

This theorem extends prior results, such as the generalization bounds established by Foret et al. (2021) and Kim et al. (2022), from Euclidean space to a broader, more general reproducing kernel Hilbert space. It is noteworthy that this is not a straightforward extension, as the previous generalization bounds rely on the dimensionality of the domain, while the RKHS is typically infinite-dimensional for many widely used kernels such as the RBF kernels (Aronszajn, 1950). Building on the first theorem, we present the second theorem, which directly addresses the population posterior and serves as the primary motivation for our method. The proof of this theorem can be found in Appendix A.3.

**Theorem 4.3** (Informal). *Let $q$ be any distribution and $d_{\mathrm{VC}}$ denotes the VC dimension of he hypothesis space $\mathcal{F} = \{f_\theta : \theta \in \Theta\}$. For any $\rho > 0$, with probability of $1 - \delta$ over the training set $\mathcal{S}$ generated by distribution $\mathcal{D}$, we have:*

$$D_{\mathrm{KL}}\left(q_{[I+f]}\|p(\theta|\mathcal{D})\right)$$
$$\leq \max_{f' \in \mathcal{H}^d, \|f'-f\| \leq \rho} D_{\mathrm{KL}}\left(q_{[I+f']}\|p(\theta|\mathcal{S})\right)$$
$$+ \mathcal{O}\left(\sqrt{\frac{\log(1 + \frac{1}{\rho^2}) + \log\left(\frac{n}{\delta}\right)}{n-1}} + \frac{\sqrt{d_{VC}\log\frac{2en}{d_{VC}}}}{\delta\sqrt{2n}}\right).$$

Our objective is to learn the function $f^* \in \mathcal{H}^d$ that minimizes $D_{\mathrm{KL}}(q_{[I+f]}\|p(\theta|\mathcal{D}))$. Motivated by Theorem 4.3, we propose to *implicitly* minimize $D_{\mathrm{KL}}(q_{[I+f]}\|p(\theta|\mathcal{D}))$ by minimizing the right-hand side term $\max_{\|f'-f\|_{\mathcal{H}^d} \leq \rho} D_{\mathrm{KL}}\left(q_{[I+f']}\|p(\theta|\mathcal{S})\right)$. For any $f \in \mathcal{H}^d$, let $F[f] = D_{\mathrm{KL}}\left(q_{[I+f]}\|p(\theta|\mathcal{S})\right)$ and $f' = f + \rho\hat{f}$, it follows that:

$$\underset{\|f'-f\|_{\mathcal{H}^d} \leq \rho}{\arg\max} D_{\mathrm{KL}}\left(q_{[I+f']}\|p(\theta|\mathcal{S})\right) \tag{7}$$

$$= \underset{\|\hat{f}\|_{\mathcal{H}^d} \leq 1}{\arg\max} F[f + \rho\hat{f}] \tag{8}$$

$$= \underset{\|\hat{f}\|_{\mathcal{H}^d} \leq 1}{\arg\max} F[f] + \rho\left\langle \hat{f}, \nabla_f F[f] \right\rangle_{\mathcal{H}^d} + \mathcal{O}(\rho^2) \tag{9}$$

$$\approx \underset{\|\hat{f}\|_{\mathcal{H}^d} \leq 1}{\arg\max} \left\langle \hat{f}, \nabla_f F[f] \right\rangle_{\mathcal{H}^d}. \tag{10}$$

Let $g = \nabla_f F[f] \in \mathcal{H}^d$. The Cauchy-Schwarz inequality on Hilbert spaces (Kreyszig, 1978) implies:

$$\left| \left\langle \hat{f}, g \right\rangle_{\mathcal{H}^d} \right| \leq \left\langle \hat{f}, g \right\rangle_{\mathcal{H}^d} \leq \|\hat{f}\|_{\mathcal{H}^d}\|g\|_{\mathcal{H}^d} \leq \|g\|_{\mathcal{H}^d}.$$

In turn, the solution $\hat{f}^*$ that solves the maximization problem in Eq. 10 is given by:

$$\hat{f}^* = \frac{g}{\|g\|_{\mathcal{H}^d}} = \frac{\nabla_f D_{\mathrm{KL}}\left(q_{[I+f]}\|p(\cdot|\mathcal{S})\right)}{\left\|\nabla_f D_{\mathrm{KL}}\left(q_{[I+f]}\|p(\cdot|\mathcal{S})\right)\right\|_{\mathcal{H}^d}}. \tag{11}$$

Recall that our goal is to find a sequence of functions $\{f_k\}_k \subset \mathcal{H}^d$ that converges toward the optimal solution $f^*$. With the sequence $\{f_k\}_k$, we can obtain the flow of distributions $\{q^{(k)}\}_k$, in which $q^{(k)} = q_{[I+f_k]}$, that gradually approaches the optimal solution of Eq. 6. Motivated by Eq. 11, we propose the following *functional sharpness-aware* update procedure:

$$\hat{f}_k^* = \rho \frac{\nabla_f D_{\mathrm{KL}}\left(q_{[I+f]}\|p(\cdot|\mathcal{S})\right)\Big|_{f=f_k}}{\left\|\nabla_f D_{\mathrm{KL}}\left(q_{[I+f]}\|p(\cdot|\mathcal{S})\right)\Big|_{f=f_k}\right\|_{\mathcal{H}^d}} \tag{12}$$

$$f_{k+1} = f_k - \epsilon\nabla_f D_{\mathrm{KL}}\left(q_{[I+f]}\|p(\cdot|\mathcal{S})\right)\Big|_{f=f_k+\hat{f}_k^*} \tag{13}$$

$$q^{(k+1)} = q_{[I+f_{k+1}]}. \tag{14}$$

To implement this iterative procedure, we must work with the functional gradient terms. For this, we rely on the following lemma, with the proof provided in Appendix B of Liu & Wang (2016):

**Lemma 4.4.** *Let $F[f] = D_{\mathrm{KL}}(q_{[I+f]} \| p(\cdot | \mathcal{S}))$. When $\|f\|$ is sufficiently small,*

$$\nabla_f F[f] \tag{15}$$
$$\approx -\mathbb{E}_q[\nabla_\theta \log p(\theta + f(\theta) | \mathcal{S}) k(\theta, \cdot) + \nabla_\theta k(\theta, \cdot)]. \tag{16}$$

Denote the right-hand side of the equation above $D(f)$. Substituting into Eq. (12) and (13), the iterative procedure described from equations (12)-(13) becomes:

$$\hat{f}_k^* = \rho \frac{D(f_k)}{\|D(f_k)\|_{\mathcal{H}^d}}, \tag{17}$$

$$f_{k+1} = f_k - \epsilon D(f_k + \hat{f}_k^*), \tag{18}$$

$$q^{(k+1)} = q_{[I+f_{k+1}]}. \tag{19}$$

Even though we do not have access to $p(\theta | \mathcal{S})$, we can compute $\nabla_\theta \log p(\theta | \mathcal{S})$ because $\nabla_\theta \log p(\theta | \mathcal{S}) = \nabla_\theta \log p(\theta) - \nabla_\theta \mathcal{L}_\mathcal{S}(\theta)$. To implement the procedure above, we first draw a set of $m$ particles $\{\theta_i^{(0)}\}_{i=1}^m$ on the model space from the initial density, and then iteratively update the particles with an empirical version of $D(f)$. Consequently, we obtain the practical procedure summarized in Algorithm 1, which deterministically transports the set of particles to match the empirical posterior distribution $p(\theta | \mathcal{S})$, therefore match the general posterior $p(\theta | \mathcal{D})$ as supported by Theorem 4.3. In Algorithm 1, at each iteration $k$, we have $m$ particles $\{\theta_j^{(k)}\}_{j=1}^m$. Eq. 17 computes the $m$ ascend steps $\hat{\varepsilon}_i^{(k)}$; then, Eq. 18 and Eq. 19 use these ascend steps to transport the $m$ model particles to $\{\theta_j^{(k+1)}\}_{j=1}^m$. It is noteworthy that FHBI is a generalization of both SVGD and SAM. In particular, if we set $\rho = 0$, we get SVGD; when $m = \#\text{PARTICLES} = 1$, we obtain SAM.

**Interactive gradient directions and Connections to SAM.** To gain further insight into the mechanism of FHBI and its connections to SAM, consider the term $\nabla_{\theta_j} \log(\theta_j + \hat{\varepsilon}_i)$ in the descending step, which is related to $\nabla_{\theta_j} \mathcal{L}_\mathcal{S}(\theta_j + \hat{\varepsilon}_i)$.

The perturbed loss can be approximated as:

$$\mathcal{L}_\mathcal{S}(\theta_j + \hat{\varepsilon}_i) \approx \mathcal{L}_\mathcal{S}(\theta_j) + \hat{\varepsilon}_i \nabla_{\theta_j} \mathcal{L}_\mathcal{S}(\theta_j),$$

where $\hat{\varepsilon}_i$ involves the average $\sum_{k=1}^m k(\theta_k, \theta_j) \nabla_{\theta_k} \mathcal{L}_\mathcal{S}(\theta_k)$. Consequently, the gradient of this perturbed loss indicates a direction that simultaneously minimizes $\|\nabla_{\theta_j} \mathcal{L}_\mathcal{S}(\theta_j)\|^2$ - which approximates the sharpness of the $j-$th particle, as discussed by Foret et al. (2021) - and $\nabla_{\theta_j} \mathcal{L}_\mathcal{S}(\theta_j) \cdot \nabla_{\theta_k} \mathcal{L}_\mathcal{S}(\theta_k)$ for all $j, k$, which reflects the angular similarity in the directions of the two particles. Thus, in addition to minimizing the sharpness of each particle, the first term of the descent step acts as an *angular repulsive force*, promoting diverse traveling directions for the particles. Besides, as discussed by Liu & Wang (2016), the second term acts as a

---

**Algorithm 1** FLAT HILBERT BAYESIAN INFERENCE (FHBI)

**Input:** Initial particles $\{\theta_i^{(0)}\}_{i=1}^m$, number of epochs $N$, step size $\rho > 0$
**Output:** A set of particles $\{\theta_i\}_{i=1}^m$ that approximates the population posterior distribution $p(\theta | \mathcal{D})$
**for** iteration $k$ **do**

$\quad \hat{\varepsilon}_i^{(k)} \leftarrow \rho \frac{\phi(\theta_i^{(k)})}{\|\phi(\theta_i^{(k)})\|}$ where $\phi(\theta) = -\frac{1}{n} \sum_{j=1}^m [k(\theta, \theta_j^{(k)}) \nabla_{\theta_j^{(k)}} \log p(\theta_j^{(k)} | \mathcal{S}) + \nabla_{\theta_j^{(k)}} k(\theta, \theta_j^{(k)})]$

$\quad \theta_i^{(k+1)} \leftarrow \theta_i^{(k)} - \epsilon_i \psi(\theta_i^{(k)}, \hat{\varepsilon}_i^{(k)})$
$\quad$ where $\psi(\theta, \varepsilon) = -\frac{1}{n} \sum_{j=1}^m [k(\theta, \theta_j^{(k)}) \nabla_{\theta_j^{(k)}} \log p(\theta_j^{(k)} + \varepsilon | \mathcal{S}) + \nabla_{\theta_j^{(k)}} k(\theta, \theta_j^{(k)})]$.

**end for**

---

*spatial repulsive force*, driving the particles apart to prevent them from collapsing into a single mode. Consequently, FHBI is not a straightforward extension of SAM to multiple independent particles; it enables the sharpness and gradient directions of the particles to interact with one another. This insight on our algorithm is summarized in Figure 1. In Section 6, we empirically demonstrate that, compared to SVGD, FHBI not only effectively minimizes particle-wise sharpness and loss values but also fosters greater diversity in the travel directions of the particles during training. This increased directional diversity, combined with the kernel gradient term, further mitigates the risk of particles collapsing into a single mode and improve the final performance as presented in Section 5.

## 5. Experiments

**Applications to Model Fine-tuning.** Bayesian inference methods have promising applications in model fine-tuning. We are given a pre-trained model $\Phi$ in standard fine-tuning scenarios. The objective is to find the optimal parameters $\theta = \Phi + \beta$, where $\beta$ represents an additional module, often lightweight and small relative to the full model. Several parameter-efficient finetuning strategies have been developed, including LoRA (Hu et al., 2022), Adapter (Houlsby et al., 2019), and others. Our experiments focus on fine-tuning the ViT-B/16 architecture (Dosovitskiy, 2021), pre-trained with the ImageNet-21K dataset (Deng et al., 2009), where $\beta$ is defined by the LoRA framework. For the Bayesian approaches, we aim to learn $m$ LoRA particles $\beta^{(i)}$ to obtain $m$ model instances $\theta^{(i)}$. The final output is then computed as the average of the outputs from all these model instances.

*Table 1.* VTAB-1K results evaluated on Top-1 accuracy. All methods are applied to finetune the same set of LoRA parameters on ViT-B/16 pre-trained with ImageNet-21K dataset.

| | Natural | | | | | | | Specialized | | | | Structured | | | | | | | | |
|---|---|---|---|---|---|---|---|---|---|---|---|---|---|---|---|---|---|---|---|---|
| Method | CIFAR100 | Caltech101 | DTD | Flower102 | Pets | SVHN | Sun397 | Camelyon | EuroSAT | Resisc45 | Retinopathy | Clevr-Count | Clevr-Dist | DMLab | KITTI | dSpr-Loc | dSpr-Ori | sNORB-Azi | sNORB-Ele | AVG |
| FFT | 68.9 | 87.7 | 64.3 | 97.2 | 86.9 | **87.4** | 38.8 | 79.7 | 95.7 | 84.2 | 73.9 | 56.3 | 58.6 | 41.7 | 65.5 | 57.5 | 46.7 | 25.7 | 29.1 | 65.6 |
| AdamW | 67.1 | 90.7 | 68.9 | 98.1 | 90.1 | 84.5 | 54.2 | 84.1 | 94.9 | 84.4 | 73.6 | 82.9 | 69.2 | 49.8 | 78.5 | **75.7** | 47.1 | 31.0 | **44.0** | 72.0 |
| SAM | 72.7 | 90.3 | 71.4 | 99.0 | 90.2 | 84.4 | 52.4 | 82.0 | 92.6 | 84.1 | 74.0 | 76.7 | 68.3 | 47.9 | 74.3 | 71.6 | 43.4 | 26.9 | 39.1 | 70.5 |
| DeepEns | 69.1 | 88.9 | 67.7 | 98.9 | 90.7 | 85.1 | 54.5 | 82.6 | 94.8 | 82.7 | 75.3 | 46.6 | 47.1 | 47.4 | 68.2 | 71.1 | 36.6 | 30.1 | 35.6 | 67.0 |
| BayesTune | 67.2 | 91.7 | 69.5 | 99.0 | 90.7 | 86.4 | 54.7 | 84.9 | **95.3** | 84.1 | 75.1 | **82.8** | 68.9 | 49.7 | 79.3 | 74.3 | 46.6 | 30.3 | 42.8 | 72.2 |
| SGLD | 68.7 | 91.0 | 67.0 | 98.6 | 89.3 | 83.0 | 51.6 | 81.2 | 93.7 | 83.2 | 76.4 | 80.0 | 70.1 | 48.2 | 76.2 | 71.1 | 39.3 | 31.2 | 38.4 | 70.4 |
| SADA-JEM | 70.3 | 91.9 | 70.2 | 98.2 | 91.2 | 85.6 | 54.7 | 84.3 | 94.1 | 83.4 | 77.0 | 79.9 | 72.1 | 51.6 | 79.4 | 70.7 | 45.3 | 29.6 | 40.1 | 72.1 |
| SA-BNN | 65.1 | 91.5 | 71.0 | 98.9 | 89.4 | 89.3 | 55.2 | 83.2 | 94.5 | 86.4 | 75.2 | 61.4 | 63.2 | 40.0 | 71.3 | 64.5 | 34.5 | 27.2 | 31.2 | 68.1 |
| SVGD | 71.3 | 90.2 | 71.0 | 98.7 | 90.2 | 84.3 | 52.7 | 83.4 | 93.2 | 86.7 | 75.1 | 75.8 | 70.7 | 49.6 | 79.9 | 69.1 | 41.2 | 30.6 | 33.1 | 70.9 |
| **FHBI** | **74.1** | **93.0** | **74.3** | **99.1** | **92.4** | 87.3 | **56.5** | **85.3** | 95.0 | **87.2** | **79.6** | 80.1 | **72.3** | **52.2** | **80.4** | 70.8 | **51.2** | **31.9** | 41.3 | **73.7** |
| | (.17) | (.42) | (.15) | (0.20) | (0.21) | (.52) | (.12) | (.31) | (.57) | (.21) | (.20) | (.16) | (.27) | (.47) | (.31) | (.50) | (.32) | (.36) | (.59) | |

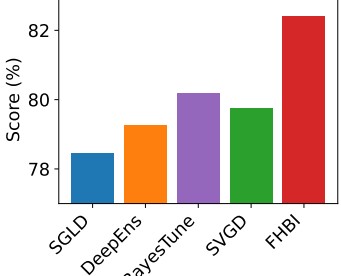
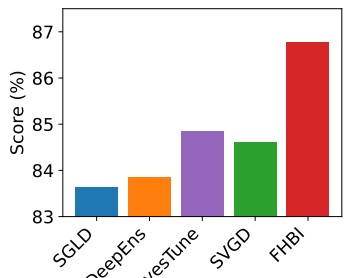
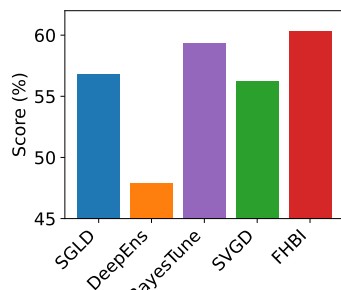

*Figure 2.* Domain-wise average scores on Natural (left), Specialized (middle), and Structured (right) datasets. FHBI performs best in all three domains compared to the Bayesian inference baselines.

**Experimental Details.** To assess the efficacy of FHBI, we experiment on VTAB-1K (Zhai et al., 2020), a challenging image classification/prediction benchmark consisting of 19 datasets from various domains. VTAB-1K covers various tasks across different semantics and object categories. The datasets are organized into Natural, Specialized, and Structured domains. We compared FHBI against nine baselines with three deterministic finetuning strategies, including full finetuning, AdamW, and SAM, and four Bayesian inference techniques, including Bayesian Deep Ensembles (Lakshminarayanan et al., 2017), BayesTune (Kim & Hospedales, 2023), Sharpness-Aware Bayesian Neural Network (SA-BNN) (Nguyen et al., 2023), Sharpness-aware Joint Energy-based Model (SADA-JEM) (Yang et al., 2023), Stochastic Gradient Langevin Dynamics (SGLD) (Welling & Teh, 2011), and Stein Variational Gradient Descent (SVGD) (Liu & Wang, 2016). The experiments were conducted on a single Tesla V100 GPU. Appendix C provides more details regarding the chosen hyperparameters.

**Experimental Results.** We present the classification accuracy results in Table 1. FHBI notably improves upon the baselines, outperforming them in most settings. Compared to other particle sampling methods, including SGLD and SVGD, FHBI consistently performs better across all settings. Moreover, FHBI improves upon SAM by a margin of 3.2%, highlighting the advantages of using multiple particles with the underlying interactive gradient directions as previously discussed in Section 4.1. Additionally, as illustrated in Figure 2, FHBI shows the highest performance across all three domains, further solidifying its advantage over the Bayesian inference baselines. Besides, FHBI also outperforms SA-BNN, which resembles the combination of SVGD and SAM, as well as SADA-JEM, which resembles SGLD and SAM, highlighting that FHBI is substantially different from a combination of SVGD and SAM as discussed in Section 4.1 and in the ablation study in Section 6.

To further assess the robustness of FHBI, we evaluate the Expected Calibration Error (ECE) of each setting. This score measures the maximum discrepancy between the model's accuracy and confidence. As indicated in Table 2, even though there is typically a trade-off between accuracy and ECE, our approach achieves a good balance between the ECE and the classification accuracy.

*Table 2.* `VTAB-1K` results evaluated on the Expected Calibration Error (ECE) metric. All methods are applied to finetune the same set of LoRA parameters on ViT-B/16 pre-trained with `ImageNet-21K` dataset.

| | Natural | | | | | | | Specialized | | | | Structured | | | | | | | | |
| --- | --- | --- | --- | --- | --- | --- | --- | --- | --- | --- | --- | --- | --- | --- | --- | --- | --- | --- | --- | --- |
| Method | CIFAR100 | Caltech101 | DTD | Flower102 | Pets | SVHN | Sun397 | Camelyon | EuroSAT | Resisc45 | Retinopathy | Clevr-Count | Clevr-Dist | DMLab | KITTI | dSpr-Loc | dSpr-Ori | sNORB-Azi | sNORB-Ele | AVG |
| FFT | 0.29 | 0.23 | 0.20 | 0.13 | 0.27 | 0.19 | 0.45 | 0.21 | 0.13 | 0.18 | 0.17 | 0.41 | 0.44 | 0.42 | 0.22 | **0.14** | 0.23 | 0.24 | 0.40 | 0.26 |
| AdamW | 0.38 | 0.19 | 0.18 | 0.05 | 0.09 | 0.10 | 0.14 | 0.11 | 0.09 | 0.12 | 0.11 | **0.12** | 0.19 | 0.34 | 0.18 | 0.14 | 0.21 | 0.18 | 0.31 | 0.17 |
| SAM | 0.21 | 0.25 | 0.20 | 0.11 | 0.12 | 0.15 | **0.14** | 0.17 | 0.16 | 0.14 | 0.09 | 0.12 | 0.17 | 0.24 | 0.16 | 0.21 | 0.19 | 0.13 | 0.16 | 0.16 |
| DeepEns | 0.24 | 0.12 | 0.22 | 0.04 | 0.10 | 0.13 | 0.23 | 0.16 | 0.07 | 0.15 | 0.21 | 0.31 | 0.32 | 0.36 | **0.13** | 0.32 | 0.31 | 0.16 | 0.29 | 0.20 |
| BayesTune | 0.32 | **0.08** | 0.20 | **0.03** | 0.85 | 0.12 | 0.22 | 0.13 | 0.07 | 0.13 | 0.22 | 0.12 | 0.23 | 0.30 | 0.24 | 0.28 | 0.28 | 0.31 | 0.26 | 0.23 |
| SGLD | 0.26 | 0.20 | 0.17 | 0.05 | 0.18 | 0.14 | 0.23 | 0.18 | 0.09 | **0.12** | 0.32 | 0.26 | 0.29 | 0.21 | 0.26 | 0.42 | 0.39 | **0.11** | 0.24 | 0.22 |
| SADA-JEM | 0.22 | 0.11 | 0.20 | 0.05 | 0.13 | 0.16 | 0.18 | 0.15 | 0.21 | 0.23 | 0.26 | 0.19 | 0.20 | 0.25 | 0.27 | 0.35 | 0.20 | 0.14 | 0.13 | 0.19 |
| SA-BNN | 0.22 | 0.08 | 0.19 | 0.15 | 0.12 | 0.12 | 0.24 | 0.13 | 0.06 | 0.12 | 0.18 | 0.14 | 0.21 | 0.22 | 0.24 | 0.25 | 0.41 | 0.46 | 0.34 | 0.20 |
| SVGD | 0.20 | 0.13 | 0.19 | 0.04 | 0.16 | 0.09 | 0.20 | 0.15 | 0.11 | 0.13 | 0.12 | 0.17 | 0.21 | 0.30 | 0.18 | 0.21 | 0.25 | 0.14 | 0.26 | 0.18 |
| **FHBI** | **0.19** | 0.10 | **0.16** | 0.06 | **0.06** | **0.09** | 0.16 | **0.09** | **0.05** | 0.12 | **0.08** | 0.14 | **0.15** | **0.21** | 0.15 | 0.16 | **0.18** | 0.11 | **0.07** | **0.12** |

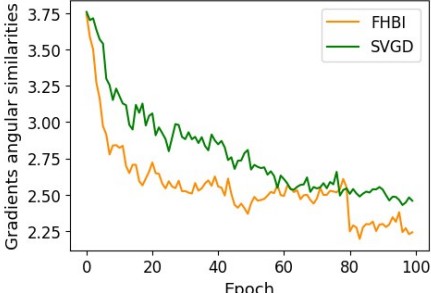

*Figure 3.* Gradients angular similarities with $m = 4$. Lower values indicate greater angular diversity.

A potential concern regarding our algorithm is the additional computational overhead compared to SVGD due to the gradient ascent step. Nevertheless, in Appendix B.1, we have conducted an experiment to assess the immpact of different number of particles. Figure 5 and 3 shows that with a negotiable tradeoff in runtime, multiple particles result in significant performance improvements compared to a single particle, and we found that `#Particles = 4` provides an optimal balance between performance gains and computational overhead for this particular application. Another potential concern is the effect of the kernel choice $k$. Indeed, we have chosen the RBF kernel due to its effectiveness in the literature of kernel methods. As we have shown in Appendix B.2, the performance gap between different kernel choices is insignificant, showing the robustness of FHBI with respect to the kernel choice.

## 6. Ablation Studies: Particles Sharpness and Gradient Diversity

As discussed in Section 4.1 and Section 5, FHBI shares implicit connections with SAM by minimizing particle-wise sharpness and diversifying travel directions, thereby

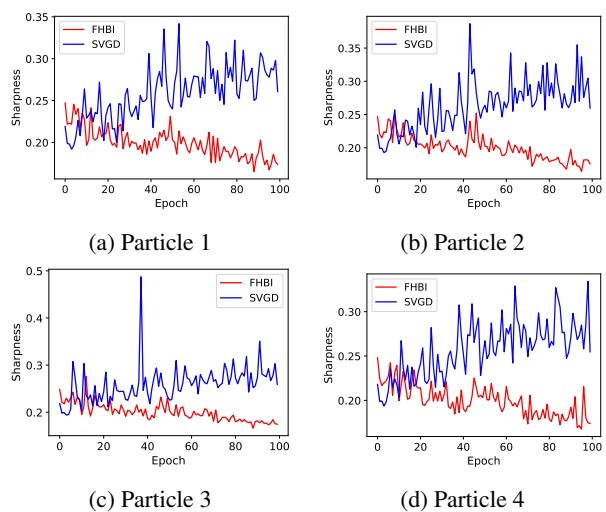

(a) Particle 1      (b) Particle 2

(c) Particle 3      (d) Particle 4

*Figure 4.* Evolution of sharpness of four particles over 100 epochs with SVGD (blue) and FHBI (red)

improving the final performance. To empirically verify this hypothesis on the behavior of our algorithm, we contrast FHBI with SVGD on the `KITTI` dataset. Four particles are initialized at the same location. We measured: **1)** the evolution of sharpness of each particle, defined as $\max_{\|\varepsilon\| \leq \rho} \mathcal{L}_{\mathcal{S}}(\boldsymbol{\theta} + \varepsilon) - \mathcal{L}_{\mathcal{S}}(\boldsymbol{\theta})$ according to Foret et al. (2021), and **2)** the evolution of gradients angular diversity, quantified as the Frobenius norm of the covariance matrix formed by the particle gradients. Figure 4 shows that FHBI results in significantly lower and more stable sharpness evolution, encouraging less congruent gradient directions and promoting particles to explore diverse trajectories. Hence, FHBI reduces particle sharpness while promoting angular diversity, thereby improving generalization ability and avoiding overfitting by collapsing into a single mode.

# 7. Conclusion

We introduce Flat Hilbert Bayesian Inference (FHBI) to enhance generalization ability beyond previous Bayesian inference approaches. This algorithm is based on a theoretical framework that extends generalization principles from Euclidean spaces to the infinite-dimensional RKHS. Empirically, FHBI consistently demonstrated significant performance improvements over nine baseline methods.

**Limitations and Future Directions.** Similar to other particle-based methods, FHBI needs to store multiple models. Although it remains well-suited for fine-tuning since the additional modules are typically lightweight, this requirement is a memory bottleneck for larger models. Given that the variational inference approaches can alleviate this issue, an avenue for future research is to extend the concept of *sharpness over functional spaces* introduced by our theory to the VI techniques to improve the generalization of these methods without storing multiple models.

# Acknowledgments

Trung Le was partly supported by ARC DP23 grant DP230101176 and by the Air Force Office of Scientific Research under award number FA9550-23-S-0001.

# Impact Statement

This paper presents work to advance the field of Machine Learning. There are potential societal consequences of our work, none of which we feel must be highlighted.

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

# Supplement to "Improving Generalization with Flat Hilbert Bayesian Inference"

The Appendix will be structured as follows: Section A presents the proofs to our theoretical results. Next, Section B is dedicated for the additional experiments to assess the effectiveness of our method, including the effect of the number of particles, and the effect of kernel choices. Then, Section C discusses the related experimental details, including the chosen hyperparameters or data augmentations used in our experiments.

## A. All Proofs

We introduce a few additional notations for the sake of the missing proofs of the main theoretical results. Given a RKHS $\mathcal{H}$ equipped with the inner product $\langle \cdot, \cdot \rangle_{\mathcal{H}}$ and the norm operator $\| \cdot \|_{\mathcal{H}}$. We define the single-sample loss function on the functional space $\mathcal{H}$ to be a map:

$$\tilde{\ell} : \mathcal{H}^d \times \mathcal{X} \times \mathcal{Y} \to \mathbb{R}$$
$$(\boldsymbol{f}, x, y) \mapsto \tilde{\ell}(\boldsymbol{f}, (x, y)).$$

Define the *general functional loss* $\tilde{L}_{\mathcal{D}}(\boldsymbol{f}) = \mathbb{E}_{(x,y)\sim\mathcal{D}}[\tilde{\ell}(\boldsymbol{f}, (x, y))]$ and the *empirical functional loss* $\tilde{L}_{\mathcal{S}}(\boldsymbol{f}) = \sum_{i=1}^{n} \tilde{\ell}(\boldsymbol{f}, (x_i, y_i))$. Throughout the proof, we assume that the parameter space is bounded by $\|\boldsymbol{\theta}\| \leq H$, the data is al bounded that $\|x\| \leq R, y \leq R$ for some $R, H \in \mathbb{R}$, and the loss function $\tilde{l}$ is $s-$Hölder continuous. Indeed, this $s-$Hölder continuity property holds for a wide range of loss functions, including the MSE loss, Huber loss, Hinge loss, Smooth L1 loss, and Logistic loss.

We introduce the following lemmas that will be used throughout the proof of our main theorems.

**Lemma A.1** (Approximation of RKHS functionals). *Let $d \in \mathbb{N}$, $\mathcal{X} = [-R, R]^K$ for some $K \in \mathbb{R}$. Consider $\mathcal{K} = \{f \in \mathcal{H} : \|f\|_{\mathcal{H}} \leq 1\}$ with $\mathcal{H}$ induced by some Mercer kernel which is $\alpha$-Holder continuous for $\alpha \in (0, 1)$ with constant $C_K \geq 0$. Suppose $F$ is $s-$Holder continuous for $s \in (0, 1]$ with constant $C_f \geq 0$. There exists some $M_0 \in \mathbb{N}$ such that for every $M \in \mathbb{N}$ with $M > M_0$, by taking some fixed $\bar{t} = \{t_i\}$ with $N \in \mathbb{N}$, we have a tanh neural network $\hat{G}$ with two hidden layers of widths at most $N(M-1)$ and $3\frac{N+1}{2}(5M)^N$ parameters satisfying*

$$\sup_{f \in \mathcal{K}} |F(f) - \hat{G}(f(\bar{t}))| \leq RC_F(\epsilon_K(\bar{t}))^s + \frac{7N^2RC_G}{M}, \tag{20}$$

*with*

$$C_G = C_F(1 + \|K[\bar{t}]^{-1}\|_{op}\sqrt{N}C_K(h_{\bar{t}})^{\alpha})^s,$$

*where $K[\bar{t}]$ is the Gram matrix of $\bar{t}$.*

*Proof.* The proof can be found in (Zhou et al., 2024) □

**Lemma A.2** (Product of RKHSs). *Given $n$ RKHSs $\mathcal{H}_1, \mathcal{H}_2, \cdots, \mathcal{H}_n$, each defined on corresponding sets $X_1, X_2, \cdots, X_n$ with kernels $k_1(x_1, y_1), \cdots k_n(x_n, y_n)$ respectively. Then, $\mathcal{H} = \bigotimes_{i=1}^{n} = \mathcal{H}_1 \times \mathcal{H}_2 \times \cdots \times \mathcal{H}_n$ is also an RKHS, with kernel $K$ that is the product of the individual kernels.*

*Proof.* The product space $\mathcal{H} = \bigotimes_{i=1}^{n} \mathcal{H}_i$ consists of tuples of functions $(f_1, f_2, \cdots, f_n)$. Firstly, we define the inner product in $\mathcal{H}$ as:

$$\langle (f_1, f_2, \cdots, f_n), (g_1, g_2, \cdots, g_n) \rangle_{\mathcal{H}} = \sum_{i=1}^{n} \langle f_i, g_i \rangle_{\mathcal{H}_i}.$$

This definition naturally defines a Hilbert space structure on $\mathcal{H}$ since each $\mathcal{H}_i$ is a Hilbert space, and the sum of inner products is linear and positive definite. Now we define the kernel for the product space:

$$k((x_1, x_2, \cdots, x_n), (y_1, y_2, \cdots, y_n)) = \prod_{i=1}^{n} k(x_i, y_i).$$

Notice that the pointwise product of positive definite kernels is a positive definite kernel, hence this kernel is valid.

We now verify the reproducing property of $\mathcal{H}$. Consider a function $f = (f_1, f_2, \cdots, f_n) \in \mathcal{H}$, and evaluate the function at a point $(x_1, x_2, \cdots, x_n) \in \bigotimes_{i=1}^{n} X_i$.

The reproducing property in each individual RKHS $\mathcal{H}_i$ implies that:

$$f_i(x_i) = \langle f_i, k_i(x_i, \cdot) \rangle_{\mathcal{H}_i}.$$

Hence, for the function $f = (f_1, \cdots, f_n)$, we get:

$$\begin{aligned}
f((x_1, x_2, \cdots, x_n)) &= (f_1(x_1), f_2(x_2), \cdots, f_n(x_n)) \\
&= (\langle f_1, k_1(x_1, \cdot) \rangle_{\mathcal{H}_1}, \langle f_2, k_2(x_2, \cdot) \rangle_{\mathcal{H}_2}, \cdots, \langle f_n, k_n(x_n, \cdot) \rangle_{\mathcal{H}_n}) \\
&= \langle (f_1, f_2, \cdots, f_n), (k_1(x_1, \cdot), k_2(x_2, \cdot), \cdots k_n(x_n, \cdot)) \rangle_{\mathcal{H}}.
\end{aligned}$$

Thus, the reproducing property holds for the product space $\mathcal{H}$. Since $\mathcal{H}$ is a Hilbert space and the kernel $k$ satisfies the reproducing property, we conclude that $\mathcal{H} = \bigotimes_{i=1}^{n} \mathcal{H}_i$ is another RKHS. $\square$

### A.1. Proof of Proposition 4.1

**Proposition A.3.** *Consider the problem of finding the distribution $\mathbb{Q}$ that solves:*

$$\mathbb{Q}^* = \min_{\mathbb{Q} \ll \mathbb{P}_\theta} \left\{ \mathbb{E}_{\theta \sim \mathbb{Q}}[\mathcal{L}_{\mathcal{D}}(\boldsymbol{\theta})] + D_{\mathrm{KL}}(\mathbb{Q} \| \mathbb{P}_\theta) \right\} \tag{21}$$

*where we search over $\mathbb{Q}$ absolutely continuos w.r.t $\mathbb{P}_{\boldsymbol{\theta}}$, and the second term is the regularization term. The closed-form solution to this problem is the **population posterior** whose density has the form:*

$$q^*(\boldsymbol{\theta}) \propto \exp(-\mathcal{L}_{\mathcal{D}}(\boldsymbol{\theta})) p(\boldsymbol{\theta}).$$

*Proof.* This proposition is the general case of **Theorem 3.1** by (Nguyen et al., 2023). Denote $q(\cdot)$ as the density function of $\mathbb{Q}$. We have:

$$\mathbb{E}_{\theta \sim \mathbb{Q}}[\mathcal{L}_{\mathcal{D}}(\boldsymbol{\theta})] + D_{\mathrm{KL}}(\mathbb{Q} \| \mathbb{P}_\theta) = \int_\Theta \mathcal{L}_{\mathcal{D}}(\boldsymbol{\theta}) q(\boldsymbol{\theta}) d\boldsymbol{\theta} + \int_\Theta q(\boldsymbol{\theta}) \log \frac{q(\boldsymbol{\theta})}{p(\boldsymbol{\theta})} d\boldsymbol{\theta}.$$

The Lagrangian is given by:

$$L(q, \alpha) = \int_\Theta \mathcal{L}_{\mathcal{D}}(\boldsymbol{\theta}) q(\boldsymbol{\theta}) d\boldsymbol{\theta} + \int_\Theta q(\boldsymbol{\theta}) \log \frac{q(\boldsymbol{\theta})}{p(\boldsymbol{\theta})} d\boldsymbol{\theta} + \alpha \left( \int q(\boldsymbol{\theta}) d\boldsymbol{\theta} - 1 \right).$$

Taking derivative with respect to $q(\boldsymbol{\theta})$, it follows

$$\begin{aligned}
\mathcal{L}_{\mathcal{D}} + \log q(\boldsymbol{\theta}) + 1 - \log p(\boldsymbol{\theta}) + \alpha &= 0, \\
q(\boldsymbol{\theta}) &= \exp(-\mathcal{L}_{\mathcal{D}}(\boldsymbol{\theta})) p(\boldsymbol{\theta}) \exp(-\alpha - 1),
\end{aligned}$$

which implies that

$$q(\boldsymbol{\theta}) \propto \exp(-\mathcal{L}_{\mathcal{D}}(\boldsymbol{\theta})) p(\boldsymbol{\theta}).$$

Then, the optimal solution is the population posterior $p(\boldsymbol{\theta}|\mathcal{S})$, which concludes the proof. $\square$

### A.2. Proof of Theorem 4.2

**Theorem A.4.** *For any $\rho > 0$ and any distribution $\mathcal{D}$, with probability $1 - \delta$ over the choice of the training set $\mathcal{S} \sim \mathcal{D}^n$,*

$$\tilde{L}_{\mathcal{D}}(\boldsymbol{f}) \leq \max_{\|\boldsymbol{f}' - \boldsymbol{f}\|_{\mathcal{H}^d} \leq \rho} \tilde{L}_{\mathcal{S}}(\boldsymbol{f}') +$$

$$+ \sqrt{\frac{N' \log\left(1 + \frac{C}{\rho^2 P^2}\left(1 + \sqrt{\frac{\log(N)}{N'}}\right)^2\right) + 4\log\frac{n}{\delta} + 8\log(6n + 3k)}{n - 1}}.$$

*Proof.* $\tilde{\ell}$ is a functional that maps from $\mathcal{H}^d \times \mathcal{X} \times \mathcal{Y}$ to $\mathbb{R}$. Notice that $\mathcal{H}^d$ is a RKHS, $\mathcal{X} = \mathbb{R}^a$ and $\mathcal{Y} = \mathbb{R}^b$ for some $a, b \in \mathbb{Z}$ are Euclidean spaces, which are also instances of RKHS. Moreover, the product of RKHS's is also a RKHS according to Lemma A.2. Hence, $\mathcal{H}^d \times \mathcal{X} \times \mathcal{Y}$ is also a RKHS. According to Lemma A.1, there exists $N$ points $\overline{\boldsymbol{\theta}} = [\boldsymbol{\theta}_i]_{i=1}^N \subset \Theta$, and a two-layer neural network $G_{\boldsymbol{W}}$ parameterized by $\boldsymbol{W}$ so that

$$|\tilde{\ell}(\boldsymbol{f}, x, y) - G_{\boldsymbol{W}}(\boldsymbol{f}(\overline{\boldsymbol{\theta}}), x, y)| \leq RC_F(\epsilon_K(\bar{t}))^s + \frac{7N^2 RC_G}{M},$$

for every $(\boldsymbol{f}, x, y) \in \mathcal{H}^d \times \mathcal{X} \times \mathcal{Y}$. Consider $\boldsymbol{f}' \in \mathcal{H}^d$ so that $\|\boldsymbol{f}' - \boldsymbol{f}\| \leq \rho$. Recall that we have $\overline{\boldsymbol{\theta}} = [\boldsymbol{\theta}_i]_{i=1}^N$ where each $\boldsymbol{\theta}_i \in \mathbb{R}^d$, and $f(\overline{\boldsymbol{\theta}}) = [f(\boldsymbol{\theta}_1), \cdots, f(\boldsymbol{\theta}_N)] \in \mathbb{R}^{N'}$ where $N' = Nd$. Then, we have:

$$\|f(\overline{\boldsymbol{\theta}}) - f'(\overline{\boldsymbol{\theta}})\|_2^2 = \left(\sum \|f(\boldsymbol{\theta}_i) - f'(\boldsymbol{\theta}_i)\|\right)^{1/2}.$$

Now let $f = [f_1, f_2, \cdots, f_d]$ and $f' = [f_1', f_2', \cdots, f_d']$ where $f_i, f_j' \in \mathcal{H}$, we have:

$$\|f(\boldsymbol{\theta}_i) - f'(\boldsymbol{\theta}_i)\| = \sum_{j=1}^d |f_j(\boldsymbol{\theta}_i) - f_j'(\boldsymbol{\theta}_i)| = \sum_{j=1}^d |\langle k(\boldsymbol{\theta}_i, \cdot), f_j - f_j'\rangle| \leq \sum_{j=1}^d \|k(\boldsymbol{\theta}_i, \cdot)\| \|f_j - f_j'\| = k(\boldsymbol{\theta}_i, \boldsymbol{\theta}_i)\|f - f'\|.$$

Therefore, it follows:

$$\|f(\overline{\boldsymbol{\theta}}) - f'(\overline{\boldsymbol{\theta}})\| \leq \left(\sum k(\boldsymbol{\theta}_i, \boldsymbol{\theta}_i)\right)^{1/2} \|f - f'\|.$$

Under the assumption that $k(\boldsymbol{\theta}, \boldsymbol{\theta}) \leq C$ (this is true for the widely used kernels, for example, $C = 1$ for the RBF kernel) it implies $|\boldsymbol{f}(\overline{\boldsymbol{\theta}}) - \boldsymbol{f}'(\overline{\boldsymbol{\theta}})| \leq P\|\boldsymbol{f} - \boldsymbol{f}'\|_{\mathcal{H}^d} \leq P\rho$ where $P = CN^{1/2}$. Denote $\tilde{\boldsymbol{\theta}} = \boldsymbol{f}(\overline{\boldsymbol{\theta}}) \in \mathbb{R}^{N'}$ for some $N' \in \mathbb{Z}$, by invoking the inequality from (Foret et al., 2021), let $\rho' = \rho P$, it follows that:

$$\tilde{L}_{\mathcal{D}}(\boldsymbol{f}) = \mathbb{E}_{(x,y)\sim\mathcal{D}}[\tilde{\ell}(\boldsymbol{f}, x, y)] \leq \mathbb{E}_{(x,y)\sim\mathcal{D}}[G_{\boldsymbol{W}}(\boldsymbol{f}(\overline{\boldsymbol{\theta}}), x, y)] + RC_F(\epsilon_K(\bar{t}))^s + \frac{7N^2 RC_G}{M}$$

$$= \mathbb{E}_{(x,y)\sim\mathcal{D}}[G_{\boldsymbol{W}}(\tilde{\boldsymbol{\theta}}, x, y)] + RC_F(\epsilon_K(\bar{t}))^s + \frac{7N^2 RC_G}{M}$$

$$\leq \max_{\|\tilde{\boldsymbol{\theta}}' - \tilde{\boldsymbol{\theta}}\|_2^2 \leq \rho'} \frac{1}{n} \sum_{i=1}^n G_{\boldsymbol{W}}(\tilde{\boldsymbol{\theta}}', x, y) + h(M, N)$$

$$+ \sqrt{\frac{N' \log\left(1 + \frac{C}{\rho'^2}\left(1 + \sqrt{\frac{\log(N)}{N'}}\right)^2\right) + 4\log\frac{n}{\delta} + 8\log(6n + 3k)}{n - 1}},$$

where we defined $h(M, N) = RC_F(\epsilon_K(\bar{t}))^s + \frac{7N^2 RC_G}{M}$. By definition, a RKHS is a closed Hilbert space. Then, there exists a sequence $\{\boldsymbol{f}_n'\}$ so that $\boldsymbol{f}_n'(\overline{\boldsymbol{\theta}})$ that gets arbitrarily close to $\tilde{\boldsymbol{\theta}}'$. Then, for any $\epsilon > 0$, it follows:

$$\tilde{L}_{\mathcal{D}}(\boldsymbol{f}) \leq \max_{\|\tilde{\boldsymbol{\theta}}' - \tilde{\boldsymbol{\theta}}\|_2^2 \leq \rho'} \frac{1}{n} \sum_{i=1}^{n} G_{\boldsymbol{W}}(\tilde{\boldsymbol{\theta}}', x, y) + h(M, N)$$

$$+ \sqrt{\frac{N' \log \left(1 + \frac{C}{\rho'^2}\left(1 + \sqrt{\frac{\log(N)}{N'}}\right)^2\right) + 4 \log \frac{n}{\delta} + 8 \log(6n + 3k)}{n - 1}}$$

$$\leq \max_{\|\boldsymbol{f}'(\overline{\boldsymbol{\theta}}) - \boldsymbol{f}(\overline{\boldsymbol{\theta}})\|_2^2 \leq \rho P} \frac{1}{n} \sum_{i=1}^{n} G_{\boldsymbol{W}}(\boldsymbol{f}'(\overline{\boldsymbol{\theta}}), x, y) + h(M, N) + \epsilon \mathcal{O}(1)$$

$$+ \sqrt{\frac{N' \log \left(1 + \frac{C}{\rho^2 P^2}\left(1 + \sqrt{\frac{\log(N)}{N'}}\right)^2\right) + 4 \log \frac{n}{\delta} + 8 \log(6n + 3k)}{n - 1}}$$

$$\leq \max_{\|\boldsymbol{f}' - \boldsymbol{f}\|_2^2 \leq \rho} \frac{1}{n} \sum_{i=1}^{n} G_{\boldsymbol{W}}(\boldsymbol{f}'(\boldsymbol{\theta}), x, y) + h(M, N) + \epsilon \mathcal{O}(1)$$

$$+ \sqrt{\frac{N' \log \left(1 + \frac{C}{\rho^2 P^2}\left(1 + \sqrt{\frac{\log(N)}{N'}}\right)^2\right) + 4 \log \frac{n}{\delta} + 8 \log(6n + 3k)}{n - 1}}$$

$$\leq \max_{\|\boldsymbol{f}' - \boldsymbol{f}\|_2^2 \leq \rho} \tilde{L}_{\mathcal{S}}(\boldsymbol{f}') + h(M, N) + \epsilon \mathcal{O}(1)$$

$$+ \sqrt{\frac{N' \log \left(1 + \frac{C}{\rho^2 P^2}\left(1 + \sqrt{\frac{\log(N)}{N'}}\right)^2\right) + 4 \log \frac{n}{\delta} + 8 \log(6n + 3k)}{n - 1}}.$$

This is true for any $\epsilon > 0$. Moreover, we can choose $\epsilon_K$ and $M$ to be arbitrarily small so that $h(M, N) \to 0$. Hence, it implies

$$\tilde{L}_{\mathcal{D}}(\boldsymbol{f}) \leq \max_{\|\boldsymbol{f}' - \boldsymbol{f}\|_2^2 \leq \rho} \tilde{L}_{\mathcal{S}}(\boldsymbol{f}') +$$

$$+ \sqrt{\frac{N' \log \left(1 + \frac{C}{\rho^2 P^2}\left(1 + \sqrt{\frac{\log(N)}{N'}}\right)^2\right) + 4 \log \frac{n}{\delta} + 8 \log(6n + 3k)}{n - 1}},$$

which concludes our proof. □

### A.3. Proof of Theorem 4.3

Now we can prove the Theorem 4.3. We restate the theorem

**Theorem A.5.** *For any target distribution $p$, reference distribution $q$, and any $\rho > 0$, we have the following bound between*

*the general KL loss and the empirical KL loss*

$$D_{\mathrm{KL}}\left(q_{[\boldsymbol{f}]}||p\left(\boldsymbol{\theta}|\mathcal{D}\right)\right) \leq \max_{\boldsymbol{f}' \in \mathcal{B}_\rho(\boldsymbol{f})} D_{\mathrm{KL}}\left(q_{[\boldsymbol{f}']}||p\left(\boldsymbol{\theta}|\mathcal{S}\right)\right)$$
$$+ \sqrt{\frac{N'\log\left(1 + \frac{C}{\rho^2 P^2}\left(1 + \sqrt{\frac{\log(N)}{N'}}\right)^2\right) + 4\log\frac{n}{\delta} + 8\log(6n + 3k)}{n-1}}.$$

*Proof.* Consider the left-hand side, we have:

$$D_{\mathrm{KL}}(q_{[\boldsymbol{f}]}||p(\boldsymbol{\theta}|\mathcal{D})) = \int q_{[\boldsymbol{f}]}(\boldsymbol{\theta})\left(\mathcal{L}_{\mathcal{D}}(\boldsymbol{\theta}) + \log\frac{q_{[\boldsymbol{f}]}(\boldsymbol{\theta})}{p(\boldsymbol{\theta})} + \log Z_{\mathcal{D}}\right)d\boldsymbol{\theta}$$

$$= \int q_{[\boldsymbol{f}]}(\boldsymbol{\theta})\left(\mathbb{E}_{(x,y)\sim\mathcal{D}}\ell(\boldsymbol{\theta}, x, y) + \log\frac{q_{[\boldsymbol{f}]}(\boldsymbol{\theta})}{p(\boldsymbol{\theta})} + \log Z_{\mathcal{D}}\right)d\boldsymbol{\theta}$$

$$= \mathbb{E}_{(x,y)\sim\mathcal{D}}\left[\int q_{[\boldsymbol{f}]}(\boldsymbol{\theta})\left(\ell(\theta; x, y) + \log\frac{q_{[\boldsymbol{f}]}(\boldsymbol{\theta})}{p(\boldsymbol{\theta})}\right)d\boldsymbol{\theta}\right] + \int q_{[\boldsymbol{f}]}(\boldsymbol{\theta})\log Z_{\mathcal{D}}d\boldsymbol{\theta},$$

where $Z_{\mathcal{D}}$ is the normalizing constant. On the other hand, we also have:

$$D_{\mathrm{KL}}(q_{[\boldsymbol{f}]}||p(\boldsymbol{\theta}|\mathcal{S})) = \int q_{[\boldsymbol{f}]}\left((\boldsymbol{\theta})\mathcal{L}_{\mathcal{S}}(\boldsymbol{\theta}) + \log\frac{q_{[\boldsymbol{f}]}(\boldsymbol{\theta})}{p(\boldsymbol{\theta})} + \log Z_{\mathcal{S}}\right)d\boldsymbol{\theta}$$

$$= \int q_{[\boldsymbol{f}]}(\boldsymbol{\theta})\left(\frac{1}{n}\sum_{i=1}^{n}\ell(\boldsymbol{\theta}, x_i, y_i) + \log\frac{q_{[\boldsymbol{f}]}(\boldsymbol{\theta})}{p(\boldsymbol{\theta})} + \log Z_{\mathcal{S}}\right)d\boldsymbol{\theta}$$

$$= \frac{1}{n}\sum_{i=1}^{n}\left[\int q_{[\boldsymbol{f}]}(\boldsymbol{\theta})\left(\ell(\theta; x_i, y_i) + \log\frac{q_{[\boldsymbol{f}]}(\boldsymbol{\theta})}{p(\boldsymbol{\theta})} + \log Z_{\mathcal{S}}\right)d\boldsymbol{\theta}\right] + \int q_{[\boldsymbol{f}]}(\boldsymbol{\theta})\log Z_{\mathcal{S}}d\boldsymbol{\theta},$$

where $Z_{\mathcal{S}}$ is the normalizing constant. We define $\tilde{L}$ to be the functional such that:

$$\tilde{L}: \mathcal{H}^d \times \mathcal{X} \times \mathcal{Y} \to \mathbb{R}$$

$$(\boldsymbol{f}, x, y) \mapsto \tilde{L}(\boldsymbol{f}, x, y) = \int q_{[\boldsymbol{f}]}(\boldsymbol{\theta})\left(\ell(\theta; x, y) + \log\frac{q_{[\boldsymbol{f}]}(\boldsymbol{\theta})}{p(\boldsymbol{\theta})}\right)d\boldsymbol{\theta}.$$

According to Theorem 4.2, we have:

$$\tilde{L}_{\mathcal{D}}(\boldsymbol{f}) \leq \max_{\|\boldsymbol{f}' - \boldsymbol{f}\|_{\mathcal{H}^d} \leq \rho} \tilde{L}_{\mathcal{S}}(\boldsymbol{f}') \tag{22}$$

$$+ \sqrt{\frac{N'\log\left(1 + \frac{C}{\rho^2 P^2}\left(1 + \sqrt{\frac{\log(N)}{N'}}\right)^2\right) + 4\log\frac{n}{\delta} + 8\log(6n + 3k)}{n-1}}. \tag{23}$$

Moreover, notice that we have $Z_{\mathcal{D}} = \mathbb{E}[\exp(\mathcal{L}_{\mathcal{D}}(\boldsymbol{\theta}))p(\boldsymbol{\theta})]$, $Z_{\mathcal{S}} = \mathbb{E}[\exp(\mathcal{L}_{\mathcal{S}}(\boldsymbol{\theta}))p(\boldsymbol{\theta})]$. Let us denote $\mathrm{VCdim}(\mathcal{F}) = d_{\mathrm{VC}}$ where $\mathcal{F} = \{f_{\boldsymbol{\theta}} : \boldsymbol{\theta} \in \Theta\}$. According to Theorem 6.11 in (Shalev-Shwartz & Ben-David, 2014) (Page 51), with the

probability at least $1 - \delta$, we have

$$\sup_{\boldsymbol{\theta} \in \Theta} |\mathcal{L}_D(\boldsymbol{\theta}) - \mathcal{L}_S(\boldsymbol{\theta})| \le \frac{4 + \sqrt{\log \tau_{\mathcal{H}}(2n)}}{\delta\sqrt{2n}}.$$

Note that the growth function $\tau_{\mathcal{H}}(2n) \le \left(\frac{2en}{d_{\text{VC}}}\right)^{d_{\text{VC}}}$, we obtain

$$\sup_{\boldsymbol{\theta} \in \Theta} |\mathcal{L}_D(\boldsymbol{\theta}) - \mathcal{L}_S(\boldsymbol{\theta})| \le \frac{4 + \sqrt{d_{\text{VC}} \log \frac{2en}{d_{\text{VC}}}}}{\delta\sqrt{2n}}.$$

Therefore, for all $\boldsymbol{\theta} \in \Theta$, we gain

$$\exp(\mathcal{L}_D(\boldsymbol{\theta})) \le \exp(\mathcal{L}_S(\boldsymbol{\theta})) \exp\left(\frac{4 + \sqrt{d_{\text{VC}} \log \frac{2en}{d_{\text{VC}}}}}{\delta\sqrt{2n}}\right).$$

It follows that

$$\int \exp(\mathcal{L}_D(\boldsymbol{\theta})) p(\boldsymbol{\theta}) d\boldsymbol{\theta} \le \exp\left(\frac{4 + \sqrt{d_{\text{VC}} \log \frac{2en}{d_{\text{VC}}}}}{\delta\sqrt{2n}}\right) \int \exp(\mathcal{L}_S(\boldsymbol{\theta})) p(\boldsymbol{\theta}) d\boldsymbol{\theta}.$$

Which is equivalent to

$$Z_D \le \exp\left(\frac{4 + \sqrt{d_{\text{VC}} \log \frac{2en}{d_{\text{VC}}}}}{\delta\sqrt{2n}}\right) Z_S.$$

Therefore, we have

$$\log Z_D \le \log Z_S + \frac{4 + \sqrt{d_{\text{VC}} \log \frac{2en}{d_{\text{VC}}}}}{\delta\sqrt{2n}}$$

Combining the Inequalities 23 and A.3, it follows that:

$$D_{\text{KL}}\left(q_{[\boldsymbol{f}]} || p(\boldsymbol{\theta}|\mathcal{D})\right) \le \max_{\|\boldsymbol{f}' - \boldsymbol{f}\|_{\mathcal{H}^d} \le \rho} D_{\text{KL}}\left(q_{[\boldsymbol{f}']} || p(\boldsymbol{\theta}|\mathcal{S})\right)$$

$$+ \sqrt{\frac{N' \log\left(1 + \frac{C}{\rho^2 P^2}\left(1 + \sqrt{\frac{\log(N)}{N'}}\right)^2\right) + 4\log\frac{n}{\delta} + 8\log(6n + 3k)}{n - 1}} + \frac{4 + \sqrt{d_{\text{VC}} \log \frac{2en}{d_{\text{VC}}}}}{\delta\sqrt{2n}}.$$

which concludes our proof. $\qquad\square$

# B. Additional Experiments

### B.1. Effect of #particles

To understand the impact of varying the number of particles, we experiment on the seven Natural datasets, reporting both accuracy and per-epoch runtime. We compared FHBI with SVGD and SAM. Figure 5 and Table 3 indicate that multiple particles result in significant performance improvements compared to a single particle. However, while increasing the number of particles enhances performance, it introduces a tradeoff regarding runtime and memory required to store the models. Based on these observations, we found that using #PARTICLES = 4 provides an optimal balance between performance gains and computational overhead.

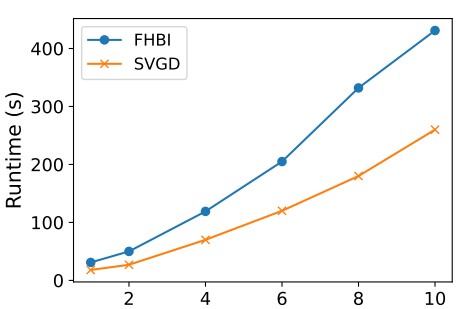

*Figure 5.* Runtime by #PARTICLES.

| #Particles | CIFAR100 | Caltech101 | DTD | Flower102 | Pets | SVHN | Sun397 |
|---|---|---|---|---|---|---|---|
| 1p (SAM) | 72.7 | 90.3 | 71.4 | 99.0 | 90.2 | 84.4 | 52.4 |
| 4p | 74.8 | 93.0 | 74.3 | **99.4** | 92.4 | 87.5 | 56.5 |
| 10p | **75.0** | **93.2** | **75.0** | 99.1 | **92.4** | **87.9** | **58.3** |

*Table 3.* Accuracy by #Particles.

### B.2. Effect of kernel choice

The implementation of FHBI relies on the choice of the kernel $k$. In our experiments, we selected the RBF kernel due to its widespread use in the kernel methods literature, known for its strong representational capabilities and its ability to balance underfitting and overfitting through the kernel width parameter $\sigma$. To evaluate the impact of different kernel choices, we tested our method on the four Specialized datasets using the polynomial kernel of degree 10 as a comparison. The results, summarized in Table 4, indicate that while the polynomial kernel slightly underperforms relative to the RBF kernel, the difference is minimal, with a performance gap of less than 0.3%.

*Table 4.* Classification accuracy on the Specialized datasets with different kernel choices

| Kernel | Camelyon | EuroSAT | Resisc45 | Retinopathy | AVG |
|---|---|---|---|---|---|
| RBF | **85.3** | **95.0** | **87.2** | **79.6** | **86.8** |
| Polynomial (d=10) | 85.0 | 94.9 | 86.8 | 79.2 | 86.5 |

### B.3. Memory consumption and scalability to larger models

To assess the scalability to larger models, we conduct an additional experiment on the larger ViT-B/16. In Tables 5 and 6, we report the memory usage and training time for the `CIFAR-100`, `Caltech-101`, and `Patch-Camelyon` datasets.

*Table 5.* VRAM consumption (MB) on ViT-B/16 and ViT-L/16

| Architechture | CIFAR-100 | Caltech-101 | Patch-Camelyon |
|---|---|---|---|
| ViT-B/16 | 12541 | 12539 | 12535 |
| ViT-L/16 | 33426 | 33684 | 32126 |

### B.4. Sensitivity to the RBF kernel length scale

As noted before, we initially tune the length scale parameter $\sigma$ from the candidates set $\{0.7, 1, 1.2\}$. To further investigate sensitivity to this hyperparameter, we expand the range to $\{0.1, 0.7, 1, 1.2, 2.5\}$, where we additionally include a small value of $\sigma = 0.1$ and a large value of $\sigma = 2.5$. The results on the `Natural` datasets, reported in Table 7, indicate that the performance remains robust within a reasonable range. We also observe a slight degradation when using extremely small values (e.g., $\sigma = 0.1$), where the model tends to overfit, or large values (e.g., $\sigma = 2.5$), where the model tends to underfit.

## C. Experimental Details

### C.1. Chosen Hyperparameters

For each experiment, we conducted five runs of FHBI and reported the mean and standard deviation. All Bayesian methods were trained with four particles on the same set of LoRA parameters. We used ten warm-up epochs, batch size 64, the Gaussian kernel, and the cosine annealing learning rate scheduler for all settings. The experiments were run with PyTorch on a Tesla V100 GPU with 40GB of RAM. FHBI involves three hyperparameters: the learning rate $\epsilon$, ascent step size $\rho$, and kernel width $\sigma$. To tune these hyperparameters, we grid-search hyperparameters on the validation set, where the key hyperparameters are: the kernel width $\sigma$, the initial learning rate $\epsilon$, and the ascent step size $\rho$. The candidate sets are formed as $\epsilon \in \{0.15, 1, 1.5, 1.7, 1.9, 2.1, 2.3, 2.5\}$, $\rho \in \{0.01, 0.03, 0.05\}$, $\sigma \in \{0.7, 1, 1.2\}$.

Table 6. VRAM consumption (MB) on ViT-B/16 and ViT-L/16

| Architechture | CIFAR-100 | Caltech-101 | Patch-Camelyon |
|---|---|---|---|
| ViT-B/16 | 5.55 | 5.66 | 5.46 |
| ViT-L/16 | 17.45 | 17.35 | 17.02 |

Table 7. Sensitivity to the RBF length scale

| $\sigma$ | CIFAR-100 | Caltech-101 | DTD | Flowers102 | Pets | SVHN | Sun397 |
|---|---|---|---|---|---|---|---|
| 0.1 | 72.1 | 91.6 | 74.0 | 97.9 | 90.2 | 85.3 | 52.2 |
| 0.7 | 73.8 | 91.8 | 73.3 | 98.7 | 92.4 | 86.7 | 56.1 |
| 1 | 73.6 | 92.7 | 72.7 | **99.1** | 91.9 | **87.3** | 54.3 |
| 1.2 | **74.1** | **93.0** | **74.3** | 98.3 | **92.4** | 86.4 | **56.5** |
| 2.5 | 69.2 | 90.9 | 69.4 | 97.5 | 90.9 | 84.6 | 52.6 |

## C.2. Data Augmentations

Our implementation is based on the repository V-PETL. Similar to this repository, we use a different data augmentation among the following three augmentations for each dataset. The data augmentations that we used for each setting are:

- For `CIFAR100, DTD, Flower102, Pets, Sun397`

```
self.transforms_train = transforms.Compose(
    [
        transforms.RandomResizedCrop(
            (self.size, self.size),
            scale=(self.min_scale, self.max_scale),
        ),
        transforms.RandomHorizontalFlip(self.flip_prob),
        transforms.TrivialAugmentWide()
        if self.use_trivial_aug
        else transforms.RandAugment(self.rand_aug_n,
                                    self.rand_aug_m),
        transforms.ToTensor(),
        transforms.Normalize(mean=[0.485, 0.456, 0.406],
                        std=[0.229, 0.224, 0.225])]),
        transforms.RandomErasing(p=self.erase_prob),
    ]
)
self.transforms_test = transforms.Compose(
    [
        transforms.Resize(
            (self.size, self.size),
        ),
        transforms.ToTensor(),
        transforms.Normalize(mean=[0.485, 0.456, 0.406],
                        std=[0.229, 0.224, 0.225])]),
    ]
)
```

- For `Caltech101, Clevr-Dist, Dsprites-Loc, Dsprites-Ori, SmallNorb-Azi, SmallNorb-Ele`:

```
self.transform_train = transforms.Compose([
    transforms.Resize((224, 224)),
    transforms.ToTensor(),
    transforms.Normalize(mean=[0.485, 0.456, 0.406],
                    std=[0.229, 0.224, 0.225])])
```

```
    self.transform_test = transforms.Compose([
        transforms.Resize((224, 224)),
        transforms.ToTensor(),
        transforms.Normalize(mean=[0.485, 0.456, 0.406],
                               std=[0.229, 0.224, 0.225])])
```

- For Clevr-Count, DMLab, EuroSAT, KITTI, Patch Camelyon, Resisc45, SVHN, Diabetic Retinopathy:

```
    from timm.data import create_transform
    self.transform_train = create_transform(
            input_size=(224, 224),
            is_training=True,
            color_jitter=0.4,
            auto_augment='rand-m9-mstd0.5-inc1',
            re_prob=0.0,
            re_mode='pixel',
            re_count=1,
            interpolation='bicubic',
        )
    aug_transform.transforms[0] = transforms.Resize((224, 224),
                                                interpolation=3)
    self.transform_test = transforms.Compose([
            transforms.Resize((224, 224)),
            transforms.ToTensor(),
            transforms.Normalize(mean=[0.485, 0.456, 0.406],
                                   std=[0.229, 0.224, 0.225])])
```

