# OpenReview forum: "Improving Generalization with Flat Hilbert Bayesian Inference"
_ICML.cc/2025/Conference — ICML 2025 poster_

### Official Review · Reviewer_nMfi · 2025-03-13

**Overall Recommendation:** 4

**Summary:**

This paper proposes a sharpness-aware version of stein variational gradient descent and applies it to neural network fine tunings.
The goal is to learn a map (in a RKHS) that transforms from the reference distribution to the true posterior by minimizing the KL divergence.
To motivate the sharpness-aware optimization for Bayesian inference, the authors prove a generalization bound, which upper bounds the true KL divergence by a worse-case KL divergence on the training data in a small neighborhood of RKHS.
Then the algorithm generally resembles the idea of Foret et al. (2021), but the difference is that the optimization is carried out in the RKHS by functional gradient descent.
The authors evaluate the propose method in neural network fine tuning, where they demonstrate improve accuracy and uncertainty estimate (measured by the ECE score).

**Claims And Evidence:**

I find the claims (both theoretical and empirical) in the paper are generally well supported.

**Essential References Not Discussed:**

Not that I am aware of.

**Experimental Designs Or Analyses:**

I read through the experiments in Section 5.

**Methods And Evaluation Criteria:**

Yes, they make sense.

**Other Comments Or Suggestions:**

1. The equations in Algorithm 1 are hard to parse of the line breaks.
Maybe use a double-column algorithm environment.

**Other Strengths And Weaknesses:**

1. The method has large memory consumption because it needs to maintain several copies of the neural network parameters.
Thus, the authors have focused on fine tuning as opposed to training from scratch.
I assume that it is very hard to apply this method to training neural networks from scratch.
But that is a common limitation of all particle based Bayesian inference methods.
1. The writing is of high quality.
1. The experiments are comprehensive across many datasets and baselines.
Experiments demonstrate that the proposed method improves not only the accuracy but also uncertainty estimates.
Additional ablation experiments demonstrate that the proposed method indeed yields more flat solutions.

**Questions For Authors:**

None.

**Relation To Broader Scientific Literature:**

This paper is a sharpness-aware variant of SVGD (Liu and Wang, 2016) by adapting the idea of SAM (Foret et al., 2021).
When applied to training neural networks, the method inherits the benefits from the both worlds---better generalization and better uncertainty estimate.

**Theoretical Claims:**

I skimmed through the proof of Theorem 4.2, but I did not check the proof of Theorem 4.3.
The authors claim that these results are non-straightforward extensions of prior results of Foret et al. (2021) because RKHS are typically infinite dimensional.
However, the proof seems to a simple combination of existing results.
They first construct a two-layer neural network that approximates the RKHS function and then the theorem is proved by invoking the results of Foret et al. (2021).

---

> ### Author Rebuttal · Authors · 2025-03-29
>
> We appreciate the reviewer’s thoughtful response. As rightly pointed out, we’ve acknowledged in the *Limitations* section that FHBI, like other particle-based Bayesian inference methods, shares the common drawback of requiring multiple models to be retained during training. This makes it less practical for training from scratch. However, FHBI remains a strong fit for fine-tuning scenarios, where the trainable components are typically lightweight. Looking ahead, an exciting direction for future work is to bring the sharpness concept in RKHS to recent variational inference (VI) methods, which do not suffer from the same memory overhead. This could offer a compelling approach in enhancing performance and robustness, while maintaining memory efficiency.

---

### Official Review · Reviewer_SGnT · 2025-03-14

**Overall Recommendation:** 4

**Summary:**

The paper introduces a novel Bayesian inference method designed to improve generalization by leveraging functional sharpness-aware particle sampling in RKHS. The key innovation lies in combining sharpness aware minimization (SAM) with SVGD in infinite-dimensional RKHS to form the proposed FHBI. The authors extend theoretical generalization bounds from finite-dimensional Euclidean spaces to infinite-dimensional functional spaces. Empirical evaluations on the VTAB-1K benchmark show that FHBI outperforms several SAM, SVGD and their combinational baselines.

## update after rebuttal
Thanks to the authors for the clarification. After reading the initial response to my questions, as well as the other reviews and replies in general, I will keep my score.

**Claims And Evidence:**

I think the claims made in the paper are clear, and supported by both theoretical proofs and experimental results.

**Essential References Not Discussed:**

I think the paper sufficiently cites prior related work.

**Experimental Designs Or Analyses:**

The experimental design is strong. It evaluates FHBI on the VTAB-1K classifications and compares it with several baselines, including SVGD, SAM-based methods, SGLD, SVGD+SAM, SGLD+SAM, and deep ensembles. Moreover, it conducts a wide ablation study about the number of particles, sharpness, and gradient diversity to further demonstrate the validity of FHBI.

**Methods And Evaluation Criteria:**

The methodology builds upon: 1) generalization theory in RKHS, extending flatness-based optimization(SAM) beyond Euclidean settings. 2) particle optimization of Stein Variational Gradient Descent (SVGD). The idea of combining SAM and SVGD in RHKS is very clear and makes sense to me.

**Other Comments Or Suggestions:**

There is a typo in the line 204: the change of variable $q(T^{-1}(\theta))$ should be $q(T^{-1}(\vartheta))$

**Other Strengths And Weaknesses:**

Strengths:
1. The idea of combining SAM with SVGD in RKHS to improve generalization is novel and generally makes sense.
2. Novel theoretical extension of generalization bounds in Euclidean space to the function space.
3. Strong empirical performance across diverse benchmarks.

Weaknesses:
1. I'm a little confused about the relationship between FHBI and SAM. It is not clearly demonstrated in the objective function.
2. Some notation is confusing, e.g., general posterior and population posterior. Are they referring to the same thing?
3. The runtime comparison in Fig. 3 only with SVGD.

**Questions For Authors:**

1. How does FHBI compare computationally to other baselines in terms of runtime and memory overhead besides SVGD?
2. As you conduct the ablation study in Sec.6 to compare the particle sharpness of FHBI and SVGD, can FHBI and SAM be compared in detail using some metric to illustrate the advantages of particle-based sampling techniques?

**Relation To Broader Scientific Literature:**

The paper is situated within the fields of Bayesian deep learning (variational methods, particle-based sampling), sharpness-aware optimization (SAM), and kernel methods in function space (RKHS).

**Theoretical Claims:**

The theoretical contributions are significant:
1. Theorem 4.2 extends Euclidean space generalization bounds to RKHS.
2. Theorem 4.3 bridges functional sharpness minimization with Bayesian inference. This establishes a connection between empirical and population KL loss.

I haven’t carefully checked every mathematical detail, but the proof appears correct.

---

> ### Author Rebuttal · Authors · 2025-03-29
>
> We appreciate the reviewer's feedback. We would like to address the concerns as follows:
>
> + **Regarding the relationship between FHBI and SAM:** As discussed at the end of Section 4, FHBI is a generalization of SAM with multiple model particles. This property is reflected more clearly in the update rules specified in the pseudocode of Algorithm 1 rather than the objective function. In particular, for the case of $m=1$ particles, the kernel terms become constant, and hence, the first step becomes the ascend step in SAM, while the second step in Algorithm 1 becomes the descend step of SAM.
>
> + **Regarding the notations:** Both terminologies in question refer to population loss. We thank the reviewer for highlighting this inconsistency and will fix this to "population loss" in the final revision to ensure consistency.
>
> + **Regarding the runtime comparison with other methods in Figure 5:** Firstly, it appears that the reviewer is referring to Figure 5 instead of Figure 3, since Figure 3 does not contain the comparisons with SVGD. Nevertheless, even though we compared FHBI with many baselines in the main experiments, we decided to ablate the runtime of FHBI by comparing it with SVGD since SVGD is most directly related to FHBI. Moreover, in Figure 5, we cannot compare with SAM because the figure presents the runtime based on the number of particles, which is not suitable for deterministic, single-particle methods like SAM.

---

### Official Review · Reviewer_JWDR · 2025-03-14

**Overall Recommendation:** 4

**Summary:**

The paper introduces Flat Hilbert Bayesian Inference (FHBI), a novel algorithm designed to enhance generalization in Bayesian inference by extending principles from finite-dimensional Euclidean spaces to infinite-dimensional reproducing kernel Hilbert spaces (RKHS). FHBI employs an iterative two-step procedure involving adversarial functional perturbation and functional descent within RKHS, supported by a theoretical framework that analyzes generalization in infinite-dimensional spaces. Empirically, FHBI is evaluated on the VTAB-1K benchmark, which includes 19 diverse datasets across various domains, where it consistently outperforms nine baseline methods by significant margins, demonstrating its practical efficacy and potential for real-world applications.

**Claims And Evidence:**

Appropriate

**Essential References Not Discussed:**

None

**Experimental Designs Or Analyses:**

The experimental designs and analyses are appropriate

**Methods And Evaluation Criteria:**

The evaluation method refers to the work of BayesTune, which I believe is acceptable in this work.

**Other Comments Or Suggestions:**

None

**Other Strengths And Weaknesses:**

1.	Although four particles were chosen as the equilibrium point, the actual training time and resource consumption (e.g., GPU memory) have not been quantified. Particularly, the scalability on large models (e.g., ViT-L/16) has not been verified.
2.	The sensitivity of the results to the length-scale of the RBF kernel was not analyzed.
3.	Although the theory proposes "functional sharpness," the experiments only indirectly assess it through empirical metrics, without designing a direct measurement method for the RKHS space.

**Questions For Authors:**

None

**Relation To Broader Scientific Literature:**

This paper is a combination of work on flat minimizers and particle-based Bayesian methods, and a detailed investigation has been conducted on the historical work of both.

**Theoretical Claims:**

All proofs except those in the supplementary materials have been checked. It can be considered that there are no significant errors.

---

> ### Author Rebuttal · Authors · 2025-03-31
>
> We thank the reviewer for the constructive feedback and would like to address the concerns as follows:
>
> + **Memory consumption and scalability with larger models:** All experiments were conducted using a single Tesla V100 GPU with 4 model particles. The memory usage and training time for the CIFAR-100, Caltech-101, and PatchCamelyon datasets are reported in the tables below. In line with the reviewer’s suggestion, we will incorporate this ablation study into the appendix of the final revision.
>
> VRAM consumption (MB):
> | Architechture | CIFAR100 | Caltech101 | Patch Camelyon |
> | ------------- | -------- | ---------- | -------------- |
> | ViT-B/16              |     12541     |     12539       |   12535             |
> | ViT-L/16          | 33426     | 33684       |      32126          |
>
> Training time (s/iter):
> | Architechture | CIFAR100 | Caltech101 | Patch Camelyon |
> | ------------- | -------- | ---------- | -------------- |
> | ViT-B/16          | 5.55     | 5.66       |      5.46          |
> | ViT-L/16              |     17.45     |    17.35      |   17.02             |
>
>
> + **Sensitivity to the RBF kernel length scale:** As noted in both the main paper and the appendix, we initially tune the length scale parameter $\sigma$ from the candidate set $\\{0.7, 1, 1.2\\}$. To further investigate sensitivity to this hyperparameter, we expand the range to $\\{0.1, 0.7, 1, 1.2, 2.5\\}$, where we additionally include a small value of $\sigma=0.1$ and a large value of $\sigma=2.5.$ The results on the Natural datasets, reported below, indicate that the performance remains robust within a reasonable range. We also observe a slight degradation when using extremely small values (e.g., $\sigma = 0.1$), where the model tends to overfit, or large values (e.g., $\sigma = 2.5$), where the model tends to underfit.
>
> | $\sigma$ | CIFAR100 | Caltech101 | DTD | Flowers102 | Pets | SVHN | Sun397 |
> | -------- | -------- | --------- |-------- |-------- |-------- |-------- |-------- |
> | 0.1     | 72.1     | 91.6     | 74.0    |97.9     |90.2     |85.3     |52.2     |
> | 0.7     | 73.8     | 91.8     | 73.3    |98.7     |92.4     |86.7     |56.1     |
> | 1       | 73.6    | 92.7    |72.7     | **99.1**     |91.9    | **87.3**     |54.3     |
> | 1.2     | **74.1**     | **93.0**        | **74.3**     |98.3    | **92.4**     |86.4     | **56.5** |
> | 2.5    | 69.2     | 90.9    |69.4    |97.5     |90.9     |84.6     |52.6    |
>
> + **Measurement of sharpness on the RKHS**: Since the transportation function $f$ governs the updates of the particles, its sharpness in the RKHS governs the sharpness of individual particles. For this reason, we decide to report the sharpness of each particle, as it more directly correlates with the model’s predictive behavior and generalization ability. As demonstrated in the ablation studies, reducing the sharpness in the RKHS leads to a corresponding reduction in the sharpness of every particle, thereby improving the generalization ability of the ensemble.

---

### Decision · Program_Chairs · 2025-05-01

**Decision:**

Accept (poster)

**Comment:**

This paper introduces Flat Hilbert Bayesian Inference (FHBI), a novel algorithm that incorporates flatness into Bayesian inference. The method combines an adversarial functional perturbation step with a functional descent step in a reproducing kernel Hilbert space (RKHS), and is theoretically grounded to improve generalization. Notably, the theoretical development is also novel, extending prior analysis of generalization from finite-dimensional Euclidean spaces to infinite-dimensional function spaces.

While the concepts of flatness and sharpness-aware minimization have been explored extensively in the community, there have been few successful attempts to incorporate flatness-seeking behavior into Bayesian posterior inference. In this regard, FHBI fills an important gap. All reviewers are positive about the novelty and significance of the contribution, and I share their assessment.